# EFFICIENT MULTIMODAL SPATIAL REASONING VIA DYNAMIC AND ASYMMETRIC ROUTING

**Yixian Shen**[1]  **Qi Bi**[1]  **Zihan Wang**[1][*]  **Zhiheng Yang**[1]  **Changshuo Wang**[2]  **Zhi Zhang**[1]
**Prayag Tiwari**[3]  **Andy D. Pimentel**[1]  **Anuj Pathania**[1][*]

[1]University of Amsterdam
{y.shen, q.bi, z.yang, z.zhang, a.d.pimentel, a.pathania}@uva.nl
zihanwang.sdu@gmail.com

[2]University College London          [3]Halmstad University
wangchangshuo1@gmail.com      prayag.tiwari@ieee.org

## ABSTRACT

Visualization-of-thought (VoT) has unlocked new capabilities for multimodal large language models (MLLMs) in complex spatial reasoning by integrating visual thinking into verbal reasoning. However, autoregressive reasoning over lengthy multimodal tokens introduces significant computation and memory overhead. We propose *DARE*, an efficient framework that adaptively prunes tokens across network depths, reasoning hops, and modalities. *DARE* employs a differentiable intra- and inter-hop retention mechanism to estimate token importance throughout reasoning, alongside an asymmetric modality-aware compression strategy that accounts for redundancy in visual–verbal representations. Moreover, a progressive KV-cache retention policy aligned with cross-modal fusion further reduces memory usage during autoregressive inference. *DARE* reduces FLOPs by 40.37% and KV-cache usage by 46.07% on average, while consistently maintaining or improving performance across seven multimodal spatial reasoning benchmarks and generalizing to broader multimodal reasoning tasks.

## 1 INTRODUCTION

Multimodal spatial reasoning involves understanding object layouts, movements, and interactions by jointly leveraging visual and linguistic cues. Such tasks often require multi-hop reasoning, where intermediate visual and textual "thoughts" are appended to the input and reprocessed in subsequent iterations. Although approaches such as VoT and MVoT are effective (Li et al., 2025a; Wu et al., 2024b), their design introduces severe scalability issues. For example, MVoT interleaves visual and verbal tokens for each thought (e.g., $32 \times 32$ tokens per image), allowing only the most recent three multimodal thoughts within a 4K context window. More broadly, the **autoregressive accumulation of intermediate tokens** rapidly increases sequence lengths, causing quadratic growth in attention cost and memory usage, which fundamentally limits current MLLMs in multi-hop spatial reasoning.

While numerous studies have explored efficient reasoning for large language models (Aytes et al., 2025; Liu et al., 2024a; Kang et al., 2025; Tan et al., 2025; Hao et al., 2024; Su et al., 2025; Cheng and Van Durme, 2024; Zhang et al., 2025a; Chen et al., 2024c; Shen et al., 2025; Wang et al., 2025; Lyu et al., 2025), these methods encounter two key challenges when applying to multi-hop multimodal spatial reasoning:

**Challenge 1: Multi-modal token importance shifts within- and across-hops.** Prior frameworks typically compress tokens within a single reasoning hop and rely on fixed or heuristic retention ratios across layers (e.g, (Chen et al., 2024b)). However, token importance varies considerably both within a hop (across network depths) and across successive hops. Specifically, as shown in Fig. 1, both visual and textual tokens follow diverse importance trajectories across layers and hops, reflecting the different semantic roles captured at each stage (e.g., objects, relations, or background context). As a result, existing models either over-retain redundant tokens or prematurely discard cues that becomes

---

[*]Corresponding Author

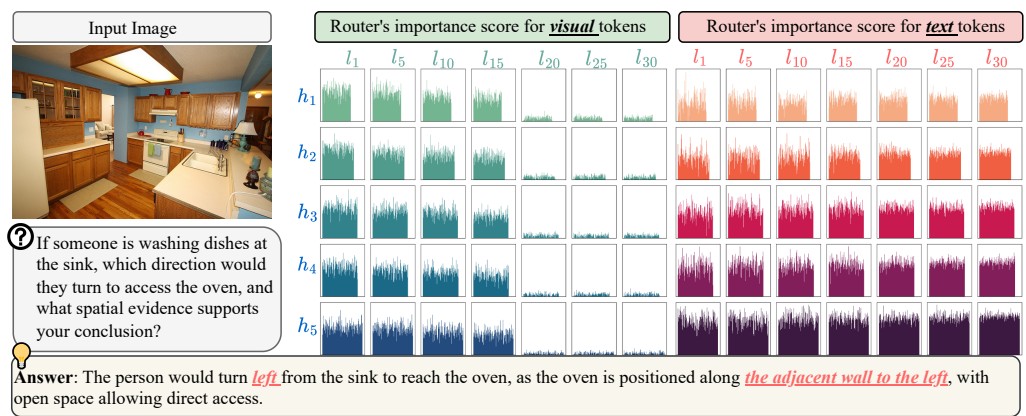

Figure 1: *DARE*'s router predictions on a spatial reasoning task. The left shows the image and question. The right figures visualize token importance scores for visual tokens (green, left grid) and text tokens (red, right grid) across 7 layers ($l_1,l_5,l_{10},l_{15},l_{20}, l_{25},l_{30}$) and 5 reasoning hops ($h_1$ to $h_5$).

critical in later reasoning steps. This gap necessitates an *intra- and inter-hop adaptive retention mechanism* that can recurrently trace token utility throughout the reasoning process.

**Challenge 2: Divergent redundancy patterns across modalities.** Existing approaches mainly target a single modality (e.g., text or images) or apply heuristic and uniform pruning across modalities. However, visual tokens exhibit distinct redundancy patterns compared to text tokens (Tan et al., 2025; Zhang et al., 2025c). As illustrated in Fig. 1, the semantic importance of visual tokens drops sharply after layer $l_{15}$, reflecting the flow of visual information into textual streams, while textual tokens remain semantically active in deeper layers. Lack of identifying and exploiting this visual-to-verbal transition leads to redundant visual retention and unnecessary computation. This gap necessitates an *asymmetric compression strategy* that aligns pruning with modality-specific redundancy patterns.

To tackle the challenges identified above, we propose a new framework, **D**ynamic and **A**symmetric **R**outing for **E**fficient multimodal spatial reasoning, named *DARE*. To address **Challenge 1**, we design an intra- and inter-hop-aware differentiable retention mechanism that adaptively estimates token utility at each network depth and reasoning step, capturing both intra-hop and inter-hop critical multimodal thoughts. To address **Challenge 2**, we introduce an asymmetric cross-modal compression strategy that independently prunes visual and textual tokens based on their distinct semantic roles and redundancy dynamics. Furthermore, motivated by the observation (see Fig. 1) that the visual token importance decays significantly after the early layers across all reasoning hops, *DARE* employs a progressive KV-cache retention strategy during inference, retaining a broader set of tokens in early layers where cross-modal fusion is most active and pruning redundant key-value entries in later stages where high-level semantic abstraction predominates.

Notably, our proposed *DARE* is fully differentiable and integrated end-to-end, allowing the model to learn optimal token retention policies during fine-tuning. Extensive experiments and ablation studies demonstrate that *DARE* substantially reduces token redundancy, computation, and memory overhead, while preserving or even enhancing reasoning performance. These results establish *DARE* as a scalable and robust recipe for efficient multi-hop multimodal reasoning. Our contributions are summarized as follows:

- We propose an ***intra- and inter-hop–aware differentiable retention mechanism*** that is fully end-to-end trainable and architecture-agnostic, enabling MLLMs to adaptively estimate token utility across depths and hops for fine-grained control.

- We develop an ***asymmetric cross-modal compression strategy*** that *leverages the visual-to-text information flow*: it prunes visual tokens more aggressively while preserving semantically critical textual tokens, aligning pruning with modality-specific redundancy dynamics.

- We introduce a ***progressive KV-cache retention policy*** that aligns with cross-modal fusion dynamics, retaining richer token sets in early layers and pruning aggressively in later stages to reduce computation and memory overhead.

- Notably, *DARE* delivers ***significant efficiency gains***, reducing FLOPs by 40.37% and KV-cache usage by 46.07% across seven multimodal spatial reasoning benchmarks, ***while consistently***

*preserving or improving accuracy*. Moreover, it **generalizes robustly** to broader reasoning tasks such as general reasoning, hallucination detection, and dialog-based VQA.

## 2 RELATED WORK

**Multimodal Spatial Reasoning (MSR).** Directly applying Chain-of-Thought (CoT) (Wei et al., 2022) to the multimodal setting poses substantial challenges in representation and efficiency (Li et al., 2024a; Wu et al., 2024a; Hurst et al., 2024). Existing efforts to improve the spatial reasoning capabilities of MLLMs generally follow three major directions: (1) *Two-stage abstraction* methods first convert visual content into intermediate symbolic representations (e.g., text (Zhang et al., 2024), graph (Mitra et al., 2024; Mondal et al., 2024), bounding boxes (Lei et al., 2024)), and are then used for downstream reasoning. (2) *Tool-augmented pipelines* integrate external components to perform reasoning over complex visual observations (Yao et al., 2023; Yang et al., 2023; Hu et al., 2024; Zhou et al., 2024; Li et al., 2024c; Gao et al., 2024). (3) *Unified sequence models* (Li et al., 2025b;a) directly interleave visual and textual tokens within the same reasoning stream, enabling fully end-to-end multimodal reasoning over multiple hops. However, such unified models produce visual and textual intermediate "thoughts" that lead to the rapidly growing sequence lengths and quadratic attention costs, limiting the reasoning depth and memory efficiency of MLLMs.

**Textual Token Compression.** Existing approaches can be broadly categorized into two types. (1) *discrete token reduction* methods aim to reduce the number of tokens via prompt engineering (Han et al., 2024; Nayab et al., 2024; Aytes et al., 2025), instruction fine-tuning (Liu et al., 2024a; Kang et al., 2025; Zhang et al., 2025a), or reinforcement learning (Arora and Zanette, 2025; Luo et al., 2025; Yang et al., 2025; Xu et al., 2025a; Mu et al., 2023). (2) *continuous latent reasoning* methods (Hao et al., 2024; Cheng and Van Durme, 2024; Deng et al., 2024; Xu et al., 2025b; Shen et al., 2025; Su et al., 2025) project intermediate reasoning steps into continuous latent spaces, which are often more efficient but suffer from less interpretability and degraded performance.

**Visual Token Compression.** As visual tokens often constitute the majority of the multimodal sequence (Tan et al., 2025) and exhibit structural redundancy, existing methods can be divided into two major paradigms. (1) *training-based methods* (Li et al., 2024b; Zhang et al., 2025b; Tong et al., 2024; Raposo et al., 2024) integrate token compression into the model architecture but overlook the visual-to-verbal information flow. (2) *training-free methods* (Chen et al., 2024c; Zhang et al., 2025c; Tan et al., 2025) compress visual tokens at inference time without modifying the model. While lightweight, these methods are limited to single-pass inference and lack hop-wise, recurrent adaptation, limiting their effectiveness and applicability for autoregressive accumulation in MSR.

## 3 METHODOLOGY

### 3.1 PRELIMINARIES

Multimodal spatial reasoning jointly interprets and reasons over multiple modalities for complex decision-making tasks. Let $\mathcal{P}_\theta$ denote a pre-trained MLLM parameterized by $\theta$, and let the input consist of textual observations $x^{(t)}$ and visual observations $x^{(v)}$. The model aims to capture both symbolic and spatial dynamics throughout the reasoning trajectory before producing the final answer. Formally, the model autoregressively generates a sequence of textual thoughts $\{t_1, \ldots, t_n\}$ and visual thoughts $\{v_1, \ldots, v_n\}$ such that:

$$\hat{v}_{i+1} \sim \mathcal{P}_\theta(v_{i+1} \mid x^{(t)}, x^{(v)}, \hat{t}_1, \hat{v}_1, \ldots, \hat{t}_i, \hat{v}_i), \quad \hat{t}_{i+1} \sim \mathcal{P}_\theta(t_{i+1} \mid x^{(t)}, x^{(v)}, \hat{t}_1, \hat{v}_1, \ldots, \hat{t}_i, \hat{v}_i), \quad (1)$$

where $\hat{t}_i$ and $\hat{v}_i$ denote the previously generated verbal and visual thoughts, respectively. The process alternates between generating a new visual thought $\hat{v}_{i+1}$ and a new textual thought $\hat{t}_{i+1}$, conditioned on the full input and the history of previous thoughts.

### 3.2 MODALITY-AWARE TOKEN ROUTING

In multi-hop multimodal spatial reasoning, the model produces long streams of visual and textual tokens, yet a large portion contributes minimally to the final prediction. Visual tokens commonly

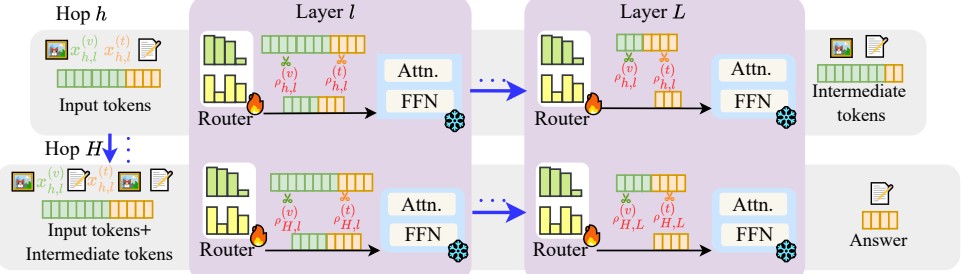

Figure 2: An overview of *DARE*. *DARE* introduces a dynamic and asymmetric token routing strategy that compresses visual $x_{h,l}^{(v)}$ and textual tokens $x_{h,l}^{(t)}$ in an intra- and inter-hop manner. Visual tokens are aggressively pruned in deeper layers, guided by the learned retention ratio $\rho_{h,l}^{(v)}$, while textual tokens are retained according to the learnable ratio $\rho_{h,l}^{(t)}$.

encode redundant background content, while textual tokens often serve only shallow syntactic purposes. These inefficiencies motivate a selective routing strategy that prioritizes semantically meaningful tokens while pruning redundant and low-utility ones.

To this end, *DARE* introduces a lightweight and modality-specific routing mechanism, as exhibited in Fig. 2, where each transformer layer integrates a modality-specific gating head to score token importance. Dedicated linear routers are designed for textual and visual embeddings, enabling fine-grained, token-level retention decisions. By pruning low-utility tokens across depths and hops, *DARE* concentrates computation on semantically and spatially critical content, significantly improving efficiency without compromising reasoning quality.

**Token Importance Prediction.** At each layer $l$, *DARE* introduces two lightweight, modality-specific routers: one for text tokens and one for visual tokens. Given the $i$-th token $x_l^{(i,m)}$ of modality $m \in \{\text{t}, \text{v}\}$, the router produces a scalar importance value through a linear projection followed by a sigmoid, which bounds it in $[0, 1]$ and makes it directly interpretable as a retention ratio.

$$s_l^{(i,m)} = \sigma\left(W_l^{(m)} x_l^{(i,m)} + b_l^{(m)}\right), \tag{2}$$

where $W_l^{(m)}$ and $b_l^{(m)}$ are learnable router parameters, and $\sigma(\cdot)$ denotes the sigmoid function. During fine-tuning, tokens retained under the target ratio $\rho_{\text{target}}^{(m)}$ have their activations scaled by the predicted importance scores. This design allows gradient signals to flow not only through the backbone layers but also into the router, enabling its parameters to be optimized end-to-end. Here, $\rho_{\text{target}}^{(m)}$ is a predefined global sparsity budget that specifies the desired retention level (see Fig. 5 for an ablation study). Formally, the processed output $y_l^{(i,m)} = \text{Layer}(x_l^{(i,m)})$ is updated as:

$$\tilde{y}_l^{(i,m)} = \alpha_l^{(i,m)} \cdot \left(s_l^{(i,m)} y_l^{(i,m)}\right) + \left(1 - \alpha_l^{(i,m)}\right) \cdot x_l^{(i,m)}, \tag{3}$$

where $\tilde{y}_l^{(i,m)}$ is the routed output to the next layer. The binary mask $\alpha_l^{(i,m)} \in \{0, 1\}$ denotes whether token $i$ falls within the top $\rho_{\text{target}}^{(m)}$ fraction of scores, with $\alpha_l^{(i,m)} = 1$ if retained and 0 otherwise.

## 3.3 Intra- and Inter-Hop Aware Retention and Asymmetric Compression

Reasoning hops and network depth emphasize different semantic cues and thus exhibit distinct token redundancy. To address this, we propose an intra- and inter-hop differentiable retention mechanism that learns modality-specific per-layer ratios $\rho_{h,l}^{(t)}$ and $\rho_{h,l}^{(v)}$, indicating the fraction of textual and visual tokens to retain. These ratios are initialized to zero and jointly optimized with model training, allowing retention to adapt dynamically across hops and depths. A global budget $\rho_{\text{target}}^{(m)}$ provides an overall constraint, while the learned ratios $\rho_{h,l}^{(m)}$ are softly regularized toward it, enabling fine-grained control within and across hops to balance retention and preserve performance (App. D.6).

We incorporate a Gumbel–Softmax relaxation (Jang et al., 2017) to approximate discrete token selection in a differentiable manner. Given token-level importance scores $s_{h,l}^{(i,m)}$, we perturb them with Gumbel noise and apply a softmax, yielding a continuous approximation of the selection mask.

$$q_{h,l}^{(i,m)} = \frac{\exp\left((s_{h,l}^{(i,m)} + g_i)/\tau\right)}{\sum_j \exp\left((s_{h,l}^{(j,m)} + g_j)/\tau\right)}, \tag{4}$$

where $g_i \sim \text{Gumbel}(0,1)$ and $\tau > 0$ is a temperature hyperparameter. As $\tau \to 0$, $q$ sharpens into a one-hot vector, collapsing to hard token selection. During training, the forward pass computes the average retention ratio $\hat{\rho}_{h,l}^{(m)}$ from $q$ and measures its gap to $\rho_{\text{target}}^{(m)}$, while a straight-through estimator treats $q$ as a surrogate in backpropagation, preserving gradient flow to the routing scores under the ratio constraint. At inference, we switch to deterministic pruning by keeping the top $\rho_{h,l}^{(m)}$ fraction of tokens based on the raw scores $s_{h,l}^{(i,m)}$. An ablation analysis of $\tau$ is provided in App. F.2.

Another key challenge in multimodal spatial reasoning lies in the structural and semantic heterogeneity between textual and visual tokens. Uniform pruning overlooks these differences, often leading to suboptimal compression. To address this, *DARE* adopts a dynamic and asymmetric routing that learns separate pruning policies for each modality. This design enables adaptive token flow control, effectively reducing redundant intermediate representations while preserving reasoning fidelity.

**Text Token Retention.** We introduce an auxiliary MSE loss over hops and layers to encourage the model to maintain a desired sparsity level for text tokens. Let $\hat{\rho}_{h,l}^{(t)}$ denote the average retention ratio of text tokens at hop $h$ and layer $l$, and let $\rho_{\text{target}}^{(t)}$ be the predefined target ratio. The loss is defined as:

$$\mathcal{L}_{\text{ratio}}^{(t)} = \frac{1}{HL} \sum_{h=1}^{H} \sum_{l=1}^{L} \left(\hat{\rho}_{h,l}^{(t)} - \rho_{\text{target}}^{(t)}\right)^2, \tag{5}$$

**Two-Phase Visual Token Retention.**

Visual tokens exhibit high spatial redundancy and contribute less to reasoning in deeper layers. To exploit this property, we introduce a two-phase retention strategy. In the *soft retention phase* (layers $l \leq l_c$), visual tokens are softly gated using importance scores $s_{h,l}^{(i,v)} \in [0,1]$ predicted by the router. We design an auxiliary MSE loss between the empirical retention ratio $\hat{\rho}_{h,l}^{(v)}$ and a fixed target $\rho_{\text{target}}^{(v)}$, encouraging the model to follow a predefined visual sparsity schedule in early layers. This loss is denoted as $\mathcal{L}_{\text{soft}}^{(v)}$ and follows the same formulation as $\mathcal{L}_{\text{ratio}}^{(t)}$ for text ones.

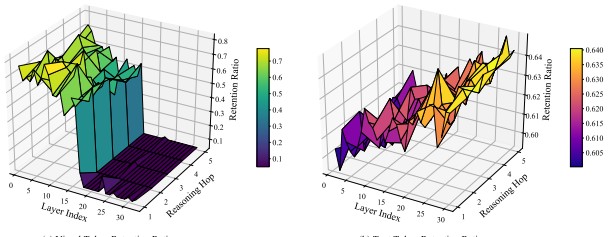

(a) Visual Token Retention Ratio    (b) Text Token Retention Ratio

Figure 3: Token retention ratios across depths and reasoning hops in *DARE*. (a) Visual token retention drops sharply in deeper layers, as spatial details lose utility once fused into language. (b) Textual token retention remains stable or rises, underscoring its increasing role in semantic reasoning. These contrasting trends motivate *DARE*'s two-phase strategy: pruning redundant visual tokens in later layers while retaining text for semantic reasoning, thus improving efficiency without loss of accuracy.

In the *hard pruning phase* (layers $l > l_c$), the cutoff layer $l_c$ is identified via a simple yet highly effective score-thresholding algorithm (e.g., the first layer where the mean visual importance falls below a threshold for two consecutive layers; see App. D.7). Once visual cues have been fused into language representations, we suppress residual activations with a penalty loss.

$$\mathcal{L}_{\text{hard}}^{(v)} = \sum_{h=1}^{H} \sum_{l=l_c+1}^{L} \sum_{i=1}^{N_{h,l}^{(v)}} \mu \cdot \max(0, s_{h,l}^{(i,v)} - \epsilon), \tag{6}$$

where $\mu$ is a penalty weight and $\epsilon$ is a small threshold (e.g., 0.01). An ablation study on the effect of $\epsilon$ is provided in App. F.1. This two-phase design retains visual tokens only when they are beneficial and prunes them aggressively in later layers, thereby significantly reducing computational overhead.

The overall training objective becomes:

$$\mathcal{L} = \mathcal{L}_{\text{task}} + \mathcal{L}_{\text{ratio}}^{(\text{t})} + \mathcal{L}_{\text{soft}}^{(\text{v})} + \mathcal{L}_{\text{hard}}^{(\text{v})}, \tag{7}$$

where $\mathcal{L}_{\text{task}}$ is the primary multimodal reasoning loss.

## 3.4 KV–CACHE RETENTION STRATEGY

*DARE* prunes tokens dynamically across depths and hops, so many tokens never produce key/value entries at layer $l$. Since pruned tokens produce no KV entries, we enrich the standard causal mask with an *execution mask* to block queries from attending to them. Formally, let $b_{h,l}^{(j,m)} \in \{0,1\}$ indicate whether token $j$ of modality $m \in \{\text{t}, \text{v}\}$ is *executed*, i.e., retained, at hop $h$ and layer $l$. In addition, we reserve a small prefix of $\kappa$ tokens per hop, validated by sensitivity analysis in Tab. 5, to (i) preserve system-level tokens such as BOS/CLS and (ii) maintain early cross-modal alignment. Because tokens from multiple hops are interleaved in the sequence, indices are defined hop-locally to avoid ambiguity. The execution mask is then defined as:

$$E_{h,l}(i,j) = \begin{cases} 0, & j \leq \kappa \text{ or } b_{h,l}^{(j,\text{t})} = 1 \text{ or } b_{h,l}^{(j,\text{v})} = 1, \\ -\infty, & \text{otherwise}, \end{cases} \tag{8}$$

and apply the composite attention mask

$$M_{h,l} = M_{\text{causal}} + E_{h,l}, \tag{9}$$

so pruned tokens receive $-\infty$ logits and are never queried. During inference, we cache $K, V$ only when $b_{h,l}^{(j,m)} = 1$; skipped tokens incur zero storage. Let $\text{mem}_{h,l}^{\text{full-t}}$ (resp. $\text{mem}_{h,l}^{\text{full-v}}$) denote the KV-cache memory that *would* be required at hop $h$, layer $l$ if *all* textual (resp. visual) tokens were cached with no pruning. The expected KV-cache memory therefore becomes

$$\mathbb{E}[\text{mem}_{h,l}] = \rho_{h,l}^{(\text{v})} \, \text{mem}_{h,l}^{\text{full-v}} + \rho_{h,l}^{(\text{t})} \, \text{mem}_{h,l}^{\text{full-t}}, \tag{10}$$

yielding substantial memory savings without degrading reasoning quality (see Tab. 4).

## 4 EXPERIMENTS

### 4.1 EXPERIMENTAL SETUP

We evaluate *DARE* on two interleaved reasoning architectures that jointly process visual and textual tokens. (1) We integrate *DARE* into VolCano(Li et al., 2025b), which interleaves text with RefBind-based visual tokens, and fine-tune it on the *VoCoT* instruction-tuning dataset using the AdamW optimizer(Loshchilov and Hutter, 2017) with a learning rate of $10^{-4}$ and weight decay of $3 \times 10^{-2}$. (2) We incorporate *DARE* into Anole-7B(Chern et al., 2024), which interleaves text with generated mental images, and fine-tune it for 60 epochs on three multi-hop spatial reasoning benchmarks: MAZE(Ivanitskiy et al., 2023), MINIBEHAVIOR(Jin et al., 2023), and FROZENLAKE(Wu et al., 2024a), following the original training hyperparameters. All experiments are run on A100 GPUs. Additional training and evaluation details are provided in the App. C. Tables highlight the best results as best and second-best as second .

**Baselines.** We compare *DARE* against a diverse set of baselines, spanning prompt-based, heuristic-based, and latent-space token compression methods. *SoT* (Aytes et al., 2025) is a static prompt-based baseline that enforces a fixed token budget through task-specific instructions. *LightFast* combines *LightThinker* (Zhang et al., 2025a) for textual token pruning with *FastV* (Chen et al., 2024c) for visual token pruning, simulating independent, modality-specific heuristics. *Heima* (Shen et al., 2025) is adapted to the multimodal setting as a latent-space reasoning baseline. *SparseVLM* (Zhang et al., 2025c) iteratively sparsifies visual tokens, retaining only the most informative ones to reduce computation. Finally, we introduce *UniPrune*, a symmetric pruning baseline that applies differentiable token scoring but omits the hard pruning loss $\mathcal{L}_{\text{hard}}^{(\text{v})}$.

**Evaluation Benchmarks.** We evaluate *DARE* across a wide range of multimodal reasoning tasks to assess both effectiveness and generality. (1) *Compositional and multi-step reasoning*, including

Table 1: Comparison on multimodal spatial reasoning. $\uparrow$ indicates higher is better; $\downarrow$ indicates lower is better. *DARE-LH* achieves competitive accuracy while significantly reducing FLOPs, latency, and memory. "-L" and "-LH" denote layer-wise and layer/hop-wise variants of *DARE*, respectively.

| Methods | $\rho_{\text{target}}^{(v)}$ | $\rho_{\text{target}}^{(t)}$ | Compositional tasks, e.g, spatial reasoning and visual search | | | | | | | |
|---|---|---|---|---|---|---|---|---|---|---|
| | | | VSR | | | | V-Star | | | |
| | | | Acc.(%)$^\uparrow$ | FLOPs(G)$^\downarrow$ | Lat.(s)$^\downarrow$ | Mem.(GB)$^\downarrow$ | Acc.(%)$^\uparrow$ | FLOPs(G)$^\downarrow$ | Lat.(s)$^\downarrow$ | Mem.(GB)$^\downarrow$ |
| VolCano | 100% | 100% | 67.18 | 19842.37 | 0.63 | 8.91 | 58.40 | 21785.41 | 0.69 | 9.37 |
| *SoT* | – | – | 53.22 | 17240.58 | 0.58 | 8.32 | 37.12 | 19820.16 | 0.64 | 8.94 |
| *LightFastV* | – | – | 57.19 | 15880.31 | 0.54 | 7.85 | 45.68 | 18340.33 | 0.59 | 8.43 |
| *SparseVLM* | – | – | 62.31 | 14827.25 | 0.49 | 7.23 | 55.95 | 17365.19 | 0.53 | 8.17 |
| *Heima* | – | – | 50.42 | **10138.96** | **0.39** | 6.14 | 40.67 | 13963.21 | **0.41** | **6.42** |
| *Unipru* | 40% | 70% | 63.71 | 14252.74 | 0.49 | 7.11 | 53.22 | 16620.27 | 0.53 | 7.69 |
| *DARE-L* | 40% | 70% | 67.13 | 12825.18 | 0.45 | 6.62 | 57.39 | 15128.02 | 0.48 | 7.14 |
| **DARE-LH** | **40%** | **70%** | **68.09** | 11310.63 | 0.41 | **6.13** | **60.07** | 13584.72 | 0.43 | 6.62 |

| Methods | $\rho_{\text{target}}^{(v)}$ | $\rho_{\text{target}}^{(t)}$ | EmbSpatial | | | | Winoground | | | |
|---|---|---|---|---|---|---|---|---|---|---|
| | | | Acc.(%)$^\uparrow$ | FLOPs(G)$^\downarrow$ | Lat.(s)$^\downarrow$ | Mem.(GB)$^\downarrow$ | Acc.(%)$^\uparrow$ | FLOPs(G)$^\downarrow$ | Lat.(s)$^\downarrow$ | Mem.(GB)$^\downarrow$ |
| VolCano | 100% | 100% | 58.29 | 26542.87 | 0.72 | 9.73 | **68.37** | 27411.36 | 0.78 | 9.84 |
| *SoT* | – | – | 54.76 | 24175.63 | 0.66 | 9.18 | 62.90 | 25984.27 | 0.72 | 9.51 |
| *LightFastV* | – | – | 56.89 | 22904.75 | 0.62 | 8.71 | 64.21 | 24632.88 | 0.67 | 8.97 |
| *SparseVLM* | – | – | 62.32 | 20275.21 | 0.72 | 8.21 | 64.39 | 23155.26 | 0.77 | 8.357 |
| *Heima* | – | – | 53.32 | **17174.83** | **0.41** | **6.92** | 64.03 | **17811.47** | **0.49** | **6.81** |
| *Unipru* | 40% | 70% | 60.75 | 21283.59 | 0.58 | 8.13 | 65.83 | 23124.63 | 0.63 | 8.43 |
| *DARE-L* | 40% | 70% | 64.37 | 19627.41 | 0.53 | 7.59 | 67.22 | 21543.90 | 0.59 | 7.94 |
| **DARE-LH** | **40%** | **70%** | **68.09** | 17964.82 | 0.45 | 7.09 | 68.31 | 19873.15 | 0.54 | 7.42 |

Table 2: Comparison on mental image generation tasks using Anole-7B across three dynamic reasoning benchmarks. We report accuracy (Acc.), compute cost (FLOPs), and inference latency (Lat.). DARE achieves the best trade-off between accuracy and efficiency.

| Model | Methods | MAZE | | | MINIBEHAVIOR | | | FROZENLAKE | | |
|---|---|---|---|---|---|---|---|---|---|---|
| | | Acc.(%)$^\uparrow$ | FLOPs(G)$^\downarrow$ | Lat.(s)$^\downarrow$ | Acc.(%)$^\uparrow$ | FLOPs(G)$^\downarrow$ | Lat.(s)$^\downarrow$ | Acc.(%)$^\uparrow$ | FLOPs(G)$^\downarrow$ | Lat.(s)$^\downarrow$ |
| | *VoT* | 86.56 | 25640.14 | 0.79 | 64.40 | 24815.83 | 0.75 | 80.21 | 23792.67 | 0.74 |
| | *MVoT* | 92.95 | 22130.56 | 0.71 | 95.14 | 21247.29 | 0.68 | 85.60 | 20591.34 | 0.67 |
| | *Heima* | 80.37 | 13243.92 | **0.42** | 68.49 | **10628.11** | 0.47 | 79.32 | **10870.96** | **0.38** |
| Anole-7B | *SparseVLM* | 87.26 | 17834.21 | 0.62 | 88.01 | 17223.23 | 0.56 | 80.29 | 17779.21 | 0.61 |
| | *Unipru* | 84.22 | 16962.41 | 0.59 | 90.34 | 16792.31 | 0.53 | 84.11 | 16225.72 | 0.57 |
| | *DARE-L* | 89.78 | 15962.41 | 0.54 | 91.25 | 14891.20 | 0.52 | 84.11 | 14072.38 | 0.50 |
| | **DARE-LH** | **93.32** | **12739.67** | 0.43 | **95.47** | 11856.21 | **0.41** | **86.11** | 11032.87 | 0.39 |

spatial reasoning (VSR (Liu et al., 2023), EmbSpatial (Du et al., 2024)), visual search (V-Star (Wu and Xie, 2024)), and Winoground (Thrush et al., 2022)). (2) *Dynamic spatial reasoning*, using MAZE(Ivanitskiy et al., 2023), MINIBEHAVIOR (Jin et al., 2023) and FROZENLAKE (Wu et al., 2024a). (3) *General VQA*, including GQA (Hudson and Manning, 2019) and MMBench (Liu et al., 2024b). (4) *Hallucination detection*, evaluated on POPE (Li et al., 2023) and AMBER (Wang et al., 2023), with CHAIR (Peng et al., 2023) used as the metric for AMBER and accuracy for the others. Additional benchmark details are provided in the App. B.

## 4.2 MAIN RESULTS

***DARE* Excels in Compositional Multi-hop Reasoning with Superior Accuracy–Efficiency Trade-off.** Tab. 1 presents the results of the VolCano model (Li et al., 2025b) across four compositional reasoning benchmarks: VSR, V-Star, EmbSpatial, and Winoground. *DARE-LH* consistently achieves the highest accuracy across all tasks while significantly reducing FLOPs, latency, and memory. On VSR and V-Star, it improves accuracy over VolCano by 0.91% and 1.67%, while reducing FLOPs by 43.0% and 37.6%, respectively. Similar efficiency gains are observed on EmbSpatial and Winoground.

Notably, while *Heima* employs latent-space reasoning for high efficiency, its accuracy lags behind due to the loss of fine-grained spatial details and explicit multimodal thought. In contrast, *DARE-LH*'s hop-aware routing selectively preserves critical spatial information, balancing accuracy and efficiency.

***DARE* Generalizes to Dynamic Spatial Reasoning with Lighter Computation.** We deploy *DARE* in Anole-7B and evaluate it on three dynamic visual reasoning benchmarks (MAZE, MINIBEHAV-

Table 3: Comparison on general QA benchmarks and hallucination benchmarks. $\downarrow$ indicates lower is better, $\uparrow$ indicates higher is better. *DARE-LH* maintains competitive accuracy while significantly reducing FLOPs, latency, and memory usage.

| Methods | $\rho_{target}^{(v)}$ | $\rho_{target}^{(t)}$ | *General VQA Tasks* | | | | | | | |
| | | | **GQA** | | | | **MMBench** | | | |
| | | | Acc.(%)$^\uparrow$ | FLOPs(G)$^\downarrow$ | Lat.(s)$^\downarrow$ | Mem.(GB)$^\downarrow$ | Acc.(%)$^\uparrow$ | FLOPs(G)$^\downarrow$ | Lat.(s)$^\downarrow$ | Mem.(GB)$^\downarrow$ |
| VolCano | 100% | 100% | 64.40 | 6052.37 | 0.30 | 6.13 | 68.11 | 7285.41 | 0.35 | 6.52 |
| *SoT* | – | – | 60.23 | 5240.58 | 0.28 | 5.76 | 63.27 | 6720.16 | 0.32 | 6.08 |
| *LightFastV* | – | – | 61.23 | 4980.31 | 0.26 | 5.28 | 63.84 | 6240.33 | 0.29 | 5.59 |
| *SparseVLM* | – | – | 61.09 | 5021.36 | 0.29 | 5.91 | 63.77 | 6492.73 | 0.31 | 5.87 |
| *Heima* | – | – | 63.42 | **3120.36** | 0.20 | **3.51** | 66.45 | **4375.63** | **0.22** | **4.12** |
| *Unipru* | 40% | 70% | 62.58 | 4352.74 | 0.24 | 4.83 | 65.12 | 5835.27 | 0.27 | 4.98 |
| **DARE-L** | **40%** | **70%** | 64.71 | 3510.63 | **0.19** | 4.21 | **68.27** | 4684.72 | **0.22** | 4.28 |

| Methods | $\rho_{target}^{(v)}$ | $\rho_{target}^{(t)}$ | *Hallucination detection tasks* | | | | | | | |
| | | | **POPE** | | | | **AMBER** | | | |
| | | | Acc.(%)$^\uparrow$ | FLOPs(G)$^\downarrow$ | Lat.(s)$^\downarrow$ | Mem.(GB)$^\downarrow$ | AMB$^\downarrow$ | FLOPs(G)$^\downarrow$ | Lat.(s)$^\downarrow$ | Mem.(GB)$^\downarrow$ |
| VolCano | 100% | 100% | 86.50 | 13285.70 | 0.43 | 6.21 | **4.60** | 13285.70 | 0.43 | 6.21 |
| *SoT* | – | – | 74.32 | 11598.12 | 0.39 | 5.82 | 6.13 | 11598.12 | 0.39 | 5.82 |
| *LightFastV* | – | – | 77.89 | 10492.24 | 0.36 | 5.38 | 5.42 | 10492.24 | 0.36 | 5.38 |
| *SparseVLM* | – | – | 78.32 | 11247.22 | 0.39 | 6.21 | 5.41 | 11764.31 | 0.37 | 5.74 |
| *Heima* | – | – | 71.38 | **6920.18** | **0.25** | **3.92** | 5.01 | **6920.18** | **0.24** | **3.92** |
| *Unipru* | 40% | 70% | 80.05 | 9272.41 | 0.33 | 4.79 | 5.12 | 9272.41 | 0.33 | 4.79 |
| **DARE-L** | **40%** | **70%** | 87.27 | 7390.41 | 0.27 | 4.27 | 4.67 | 7390.41 | 0.25 | 4.27 |

Table 4: KV-cache memory usage across spatial reasoning benchmarks. '-': Not reported.

| Method | VSR | V-Star | EmbSpatial | Winoground | MAZE | MINIBEHAVIOR | FROZENLAKE |
|---|---|---|---|---|---|---|---|
| *VoCoT* | 5.45 GB | 5.71 GB | 5.92 GB | 5.96 GB | - | - | - |
| *MVoT* | - | - | - | - | 4.61 GB | 4.52 GB | 4.46 GB |
| *DARE-L* | 3.28 GB | 3.52 GB | 3.76 GB | 3.83 GB | 2.49 GB | 2.40 GB | 2.34 GB |
| **DARE-LH** | **2.97GB** | **3.28GB** | **3.52GB** | **3.60GB** | **2.28GB** | **2.21GB** | **2.13GB** |

IOR, and FROZENLAKE) to assess its architectural generalization and adaptability to dynamic tasks. As shown in Tab. 2, *DARE-LH* achieves the highest accuracy across all benchmarks while reducing FLOPs by 40–50% and lowering latency compared to *MVoT* and *VoT*. Notably, it cuts compute on MAZE from 25.6K GFLOPs (*VoT*) to 12.7K, while improving accuracy by 6.8%.

***DARE* Preserves Accuracy with Lighter Computation on General VQA Benchmarks.** We evaluate *DARE-L* on GQA and MMBench and observe consistent gains in both accuracy and efficiency. *DARE-L* achieves the highest accuracy, matching or surpassing the full-token VolCano baseline while reducing FLOPs by 42%, latency by 30%, and memory by over 1.8 GB. Compared to *LightFastV*, which is efficient but 3.5% less accurate on GQA, *DARE* preserves performance while maintaining low compute cost, highlighting significant token redundancy even in standard benchmarks.

***DARE* Mitigates Hallucination in VLMs.** We evaluate *DARE* on POPE and AMBER to assess its ability to mitigate hallucinations in vision-language models. Tab. 3 shows *DARE-L* achieves the highest accuracy on POPE (87.27%) and the second-best hallucination rate on AMBER (AMB = 4.67). While VolCano maintains competitive accuracy, it incurs substantially higher FLOPs and memory costs. Compared to *Heima*, which relies on latent reuse, *DARE* delivers a +11.9% gain on POPE and reduces hallucinations by 0.25 on AMBER, while maintaining comparable or superior efficiency. These results demonstrate that *DARE* not only compresses effectively but also preserves factual consistency by adaptively retaining cross-modal cues essential for grounded reasoning.

**KV Cache Efficiency of *DARE*.** We analyze KV-cache usage, a dominant contributor to GPU memory during autoregressive multi-hop spatial reasoning. Tab. 4 shows that *VoCoT* and *MVoT* store all intermediate tokens within and across hops, resulting in substantial KV-cache overhead. In contrast, *DARE-LH* employs intra- and inter-hop adaptive routing, consistently achieving the lowest KV memory across all seven spatial benchmarks and reducing cache usage by over 40% on average.

**Adaptive Token Accumulation in *DARE* Mitigates Context Saturation.** Fig. 4 visualizes the temporal token usage across four representative tasks, revealing stark differences in intermediate token accumulation behavior across models. VolCano immediately saturates the available context

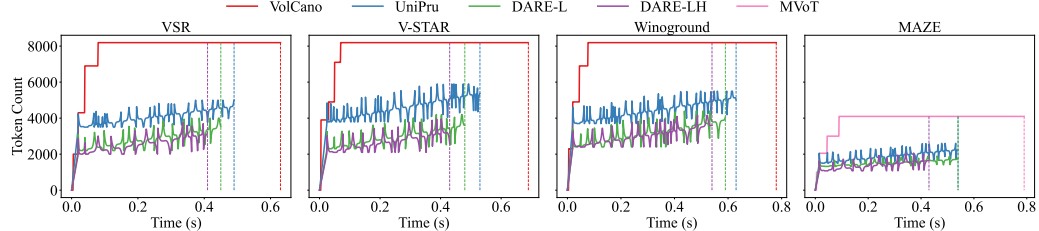

Figure 4: Token accumulation dynamics across four multimodal reasoning tasks. *DARE-LH* grows more gradually, keeping both fluctuations and token volume better controlled across hops, thereby facilitating the retention of critical information under constrained context length.

with maximal tokens (8192 or 4096), sustaining a full context limitation throughout execution. In contrast, both *DARE* variants show gradual and adaptive accumulation. *DARE-L* and *DARE-LH* exhibit a slower growth pattern, with both fluctuation and token volume regulated more conservatively across hops, saving 60% tokens compared with VolCano and *MVoT*.

## 4.3 ABLATION STUDIES

**Impact of $\mathcal{L}_{\mathbf{hard}}^{\mathbf{(v)}}$ Loss.** The $\mathcal{L}_{\text{hard}}^{(v)}$ objective promotes aggressive pruning of low-utility visual tokens in deeper layers without sacrificing performance. Tab. 1 shows that this strategy consistently reduces latency, GPU memory usage, and overall computational cost. Statistical analysis using a Student's t-test reveals that *DARE-LH* and *DARE-L* significantly outperform UniPru in efficiency, with p-values of 0.0003% and 0.0007%, respectively. These results provide strong evidence that $\mathcal{L}_{\text{hard}}^{(v)}$ enhances token efficiency while maintaining or improving accuracy.

**Token Retention Trade-Off.** We assess the impact of varying modality-specific token retention ratios on performance. Fig. 5 shows that accuracy peaks when retaining 70% of textual tokens and 40% of visual tokens. Visual tokens are more tolerant to aggressive pruning, particularly in deeper layers where their semantic contributions diminish due to cross-modal fusion. In contrast, textual tokens are more sensitive to compression, as they likely form the core of the reasoning process and are essential for maintaining both semantic and syntactic integrity.

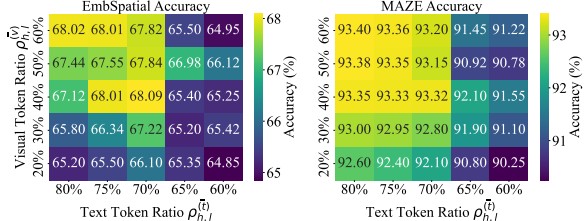

Figure 5: Ablation on modality-specific token retention.

**Impact of the prefix size $\kappa$.** *DARE* reserves a fixed prefix of $\kappa$ tokens per hop to ensure (i) system-level special tokens (e.g., BOS/CLS) are retained and (ii) early cross-modal alignment cues remain accessible under aggressive pruning. We ablate $\kappa \in 0, 1, 2, 4, 8, 16, 32$ on VSR and MAZE. Tab. 5 shows that when $\kappa = 0$ or 1, accuracy drops by 1.45–4.02%, and BOS token masking occasionally causes decoding failures. Increasing $\kappa$ beyond 2 yields negligible gains (<0.1%) while increasing KV memory by up to 70%. We find $\kappa = 2$ to be the optimal choice: it fully recovers performance, preserves key initialization tokens, and achieves KV cache efficiency.

Table 5: Impact of prefix size $\kappa$ on accuracy (%) and KV cache (GB).

| $\kappa$ | VSR | | MAZE | |
|---|---|---|---|---|
| | Acc.↑ | KV↓ | Acc.↑ | KV↓ |
| 0 | 64.12 | **2.26** | 89.30 | **1.38** |
| 1 | 65.55 | 2.60 | 91.88 | 1.84 |
| 2 | **68.09** | 2.97 | 93.32 | 2.28 |
| 4 | **68.09** | 3.40 | 93.32 | 2.62 |
| 8 | 68.07 | 3.85 | 93.32 | 3.05 |
| 16 | 68.05 | 4.35 | 93.31 | 3.55 |
| 32 | 68.06 | 4.85 | **93.33** | 4.05 |

## 5 CONCLUSION

This paper introduced *DARE*, a dynamic and asymmetric token routing framework for efficient multimodal spatial reasoning. *DARE* learns modality-, intra- and inter-hop-aware retention strategies to adaptively prune visual and textual tokens based on their evolving importance across the reasoning process. This approach enables scalable, interpretable, and resource-efficient multi-hop reasoning. Extensive experiments across diverse spatial reasoning benchmarks show that *DARE* significantly

reduces computational and memory costs, cutting FLOPs and KV-cache usage by over 40% while maintaining or even improving task performance. Beyond efficiency, its progressive KV-cache management alleviates context saturation, enabling longer-horizon multimodal reasoning. These results establish *DARE* as a scalable and robust framework for efficient multimodal reasoning.

## ACKNOWLEDGMENT

This work was funded under the COIN-3D project, which has received funding from the European Union's Horizon Europe research and innovation program under grant agreement No. 101159667.

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

CONTENTS

# A  STATEMENTS

## A.1  LLM USAGE STATEMENT

LLMs were used for language polishing and formatting. Specifically: (i) to shorten sentences, refine grammar, and improve readability (e.g., compressing section summaries, rewriting figure captions, and smoothing phrasing in the scalability and generalization sections); (ii) to provide guidance on LaTeX formatting adjustments in Overleaf, such as tuning `wrapfigure` spacing with `\vspace` and line height options; and (iii) to brainstorm alternative titles for *DARE*. The authors take full responsibility for all ideas, methods, and claims presented in this paper.

## A.2  ETHICS STATEMENT

This work does not involve human subjects, personally identifiable information, or practices that could raise ethical concerns. There are no potential conflicts of interest, and no sensitive or harmful methodologies were employed. We adhered to the ICLR Code of Ethics.

## A.3  REPRODUCIBILITY STATEMENT

We have taken measures to ensure reproducibility. Key code components are provided, and dataset descriptions, hyperparameter settings, and training procedures are detailed in the appendix and supplementary material.

# B  DATASET DETAILS

## B.1  SPATIAL REASONING BENCHMARKS.

We evaluate *DARE* across four spatial reasoning benchmarks that assess the model's ability to understand object configurations, spatial relationships, and fine-grained grounding across vision and language.

**VSR** (Liu et al., 2023) (Visual Spatial Reasoning) consists of image-text pairs where each sample includes a factual claim about the image. The model must decide whether the claim is supported by the visual evidence, making it a binary classification task. We use the *unseen test split* to measure generalization in a zero-shot setting. Following prior work (Li et al., 2025b), we format the input using the prompt: *"Is there an event {description} in the image?"* to contextualize the claim in a question-like form that encourages explicit grounding.

**EmbSpatial** (Du et al., 2024) focuses on embodied spatial understanding and includes visually complex layouts described with spatial expressions. Each question involves reasoning over fine-grained spatial relations between entities (e.g., "the red box to the left of the blue ball"). We use the official test split and preserve the original visual-textual inputs to ensure a faithful evaluation of grounding precision.

**Winoground** (Thrush et al., 2022) tests multimodal compositionality via challenging image-text alignment tasks. Each sample includes two images and two captions that differ subtly in word order or semantics (e.g., "a person holding a ball" vs. "a ball holding a person"). The model must match each image to the correct caption. We cast this as a caption selection task using the prompt: *"Please describe the image."*, which encourages the model to choose the caption that most accurately describes each visual scene.

**V-Star** (Wu and Xie, 2024) is a benchmark for visual search in cluttered scenes, requiring models to locate specific visual concepts given descriptive queries. It evaluates fine-grained recognition, disambiguation, and cross-instance grounding in complex environments. We follow the standard protocol and use the benchmark as-is without additional prompting.

In short, these datasets offer a comprehensive evaluation of spatial reasoning under diverse settings, from binary judgment and relational grounding to compositional alignment and object-level retrieval. They serve as the primary testbed for assessing the effectiveness of *DARE*'s intra- and inter-hop-aware token pruning in structured multimodal reasoning.

Table 6: Overview of spatial reasoning benchmarks.

| Category | VSR | EmbSpat. | V-Star | Wino |
|---|---|---|---|---|
| **Split** | test unseen | test | – | test |
| **Size** | 1222 | 3625 | 238 | 800 |

## B.2 DYNAMIC SPATIAL REASONING TASKS.

We evaluate *DARE* on three dynamic spatial reasoning benchmarks, namely, MAZE (Ivanitskiy et al., 2023), MINIBEHAVIOR (Jin et al., 2023), and FROZENLAKE (Wu et al., 2024a). Each dataset requires multi-step visual reasoning over simulated environments. These datasets test the model's ability to understand evolving spatial configurations, action trajectories, and implicit goals in low-level visual domains.

**MAZE** is constructed using the Maze-Dataset framework (Ivanitskiy et al., 2023), which generates 2D grid mazes via an iterative depth-first search algorithm. Mazes of size 3 to 6 are generated using multiple random seeds to diversify the layout complexity. For each instance, a navigation path is constructed, and redundant or repeated paths are filtered out to minimize knowledge leakage between training and test splits. At test time, each input consists of a maze configuration and three destination candidates (i.e., coordinate points), among which the model must select the correct goal cell. This setup emphasizes long-horizon spatial reasoning over visual layouts with minimal linguistic input.

**MINIBEHAVIOR** (Jin et al., 2023) is derived from the INSTALLINGAPRINTER simulation suite, where reinforcement learning (RL) agents are trained to complete procedural tasks in 7×7 to 10×10 grid environments using the Stable-Baselines3 library. The dataset contains diverse agent trajectories, and only successful action sequences (i.e., those completing the simulated printer installation task) are retained. The dataset applies controlled environment perturbations to prevent memorization,. Specifically, for repeated action paths or previously encountered environments, there is a 40% probability of perturbing either the printer or table coordinates, and a 20% probability of removing one of these objects. After perturbation, the agent's action sequence is replayed in the modified environment to confirm its validity. This design encourages the model to generalize reasoning across near-duplicate yet semantically different scenes,

**FROZENLAKE** (Wu et al., 2024a) is adapted from OpenAI Gym's FrozenLake environment. It consists of grid-based navigation tasks where an agent must reach a goal while avoiding holes, using Q-table-based policies to guide action selection. Trajectories are generated from agents acting greedily with respect to Q-values. Successful action sequences are included only if they haven't been seen before in the same environment. For unsuccessful attempts (e.g., falling into a hole), the trajectory is included with 50% probability either in its original form or with appended random actions. In ambiguous cases (e.g., the agent neither fails nor succeeds), the trajectory is retained to increase coverage. Additionally, Q-tables are re-learned with randomly perturbed reward paths to introduce variance and avoid overfitting. The resulting benchmark challenges models to reason over uncertain, sparse-reward settings where the structure of the environment must be inferred through action sequences.

We follow the same experimental conditions as MVoT (Li et al., 2025a). Table 7 summarizes the core characteristics of the three dynamic spatial reasoning benchmarks used to evaluate *DARE*: MAZE, MINIBEHAVIOR, and FROZENLAKE. These benchmarks differ in grid sizes, entity types, and action sequence complexity, capturing a range of spatial-temporal reasoning challenges. **MAZE** features procedurally generated layouts with deterministic navigation paths and moderate action diversity. **MINIBEHAVIOR** introduces greater behavioral complexity, offering a larger action space and controlled environment perturbations that test generalization over procedural tasks. **FROZEN-LAKE**, by contrast, is defined by its stochastic transitions, variable action lengths, and explicit reasoning over patterned environments (e.g., slippery tiles and traps), making it the most structurally demanding. Notably, only FROZENLAKE includes explicit pattern modeling, as indicated in the "Pattern Details" row. Reported action lengths and entity counts are averaged across samples to reflect typical task complexity. All three datasets provide comparably sized training and testing splits, supporting controlled evaluations of *DARE*'s spatial-temporal token retention strategies. Overall,

Table 7: Characteristics of dynamic spatial reasoning tasks, highlighting varying complexities in action dynamics and structural patterns.

| Task | MAZE | MINIBEHAVIOR | FROZENLAKE |
|---|---|---|---|
| Grid Sizes | 3–6 | 5–8 | 3–6 |
| Entity Types | 5 | 3 | 3 |
| Entities Numbers | 5 | 3 | 7.16 |
| Action Length | 9.11 | 7.83 | 6.56 |
| Action Types | 4 | 7 | 4 |
| Pattern Details | ✗ | ✗ | ✓ |
| Train Set Size | 5007 | 6400 | 6846 |
| Test Set Size | 1255 | 1604 | 1664 |

Table 8: Overview of General VQA Benchmarks and Hallucination Benchmarks

| Category | General VQA | | Hallucination | |
|---|---|---|---|---|
| Dataset | GQA | MMBench | POPE | AMBER |
| Split | testdev_balanced | DEV | adversarial | generative |
| Size | 12578 | 4329 | 3000 | 1004 |

these benchmarks form a diverse testbed that complements static spatial reasoning tasks, emphasizing trajectory grounding, goal-directed prediction, and dynamic decision-path comprehension.

### B.3 GENERAL VQA BENCHMARKS.

We evaluate *DARE* on two widely used vision-language benchmarks to assess its generalization beyond spatial reasoning: GQA (Hudson and Manning, 2019) and MMBench (Liu et al., 2024b).

For GQA, we follow prior work (Liu et al., 2023) and use the "testdev_balanced" split, which provides a balanced distribution over question types and supports rigorous evaluation of compositional reasoning and object-centric grounding. We prepend the instruction prompt: *"Please visualize the answer if you are not sure about the details."* in order to promote concise and confident outputs. This encourages the model to rely more explicitly on visual content when answering.

For MMBench, we adopt the official "DEV" split for efficient evaluation. MMBench is a manually curated multi-dimensional benchmark that tests model capabilities across 12 skill categories, including object recognition, attribute understanding, spatial relations, and commonsense reasoning. It emphasizes fine-grained vision-language alignment and has been used as a standard diagnostic tool for evaluating VLM performance under diverse visual and linguistic challenges.

Overall, these two benchmarks offer complementary perspectives on general vision-language understanding and help validate the effectiveness of *DARE*'s compression mechanism beyond its primary spatial reasoning setup.

### B.4 HALLUCINATION BENCHMARKS.

We assess the factual reliability and grounding capability of *DARE* by benchmarking it on two recent hallucination detection datasets: POPE (Li et al., 2023) and AMBER (Wang et al., 2023). These benchmarks evaluate whether a model generates content that is not supported or entailed by the visual input.

**POPE** (Perception-Oriented Probe for Evaluation) (Li et al., 2023) introduces a suite of contrastive visual examples designed to probe factual grounding and resistance to object hallucination. We follow prior work (Li et al., 2025b) and evaluate on the *adversarial subset*, which contains the most challenging examples where visual distractors are introduced to provoke hallucinations. Each sample presents a claim about the image (e.g., "There is a cat on the table"), and the model is asked to verify its correctness. We adopt the binary yes-or-no prompting protocol from the original dataset,

ensuring consistency with the official evaluation setup. This setting stresses the model's ability to avoid overconfident assertions unsupported by visual evidence.

**AMBER** (Wang et al., 2023) (A Benchmark for Evaluating Realistic Hallucinations in Multimodal Models) tests the model's ability to generate grounded, factual descriptions of images in free-form language. It consists of diverse images with multiple levels of hallucination risk and uses CHAIR (Peng et al., 2023) as the underlying hallucination scoring metric. We evaluate *DARE* on the generative task split, where the model must produce a description for each image. The original prompts from the dataset are preserved to maintain fair comparison. Generated responses are assessed for consistency, relevance, and hallucination rate using the CHAIR metric, which quantifies object-level mismatches between the output and the ground truth.

These two benchmarks offer complementary views of hallucination: POPE provides a focused and controllable binary setting, while AMBER offers open-ended generation under real-world uncertainty. In a nutshell, they allow us to assess whether *DARE*'s sparsity-aware routing improves not only efficiency but also the factual alignment of multimodal outputs.

## B.5 VISION–LANGUAGE REASONING DATASETS

NLVR2 (Suhr et al., 2018) is a benchmark dataset designed to evaluate models' ability to perform compositional reasoning over images paired with text. Each example presents a natural photograph and a pair of human-written captions that are closely related but differ in subtle semantic aspects, requiring fine-grained visual and linguistic understanding to determine which caption is true of the image. Unlike simple recognition tasks, this dataset emphasizes relational and contextual reasoning, such as spatial relations, object attributes, and logical consistency between image and language.

VLEP (Video-and-Language Event Prediction) (Lei et al., 2020) is a large-scale dataset introduced to study future event prediction in multimodal settings. Each example consists of a short video clip paired with a natural language description of the observed context, followed by two candidate textual hypotheses about what is more likely to happen next. Models must select the correct continuation, requiring them to integrate temporal visual cues with linguistic semantics and commonsense knowledge. Covering diverse everyday scenarios, VLEP emphasizes anticipatory reasoning beyond recognition, making it a challenging benchmark for evaluating video–language models' ability to understand events and predict plausible outcomes.

## B.6 DIALOG VQA

CLEVR-Dialog (Kottur et al., 2019) is a synthetic diagnostic dataset designed to evaluate multi-round reasoning in visual dialog systems. Built on the CLEVR framework, it pairs images of 3D-rendered objects with automatically generated multi-turn question–answer dialogues that probe complex reasoning skills such as counting, attribute comparison, spatial relations, and coreference resolution. Each dialogue requires the model to maintain contextual memory across rounds while grounding linguistic references in the visual scene. By controlling the visual environment and dialogue generation process, CLEVR-Dialog provides a clean and interpretable benchmark for studying compositional, multi-step reasoning in vision–language interaction.

## C IMPLEMENTATION AND HYPERPARAMETER SETTINGS

### C.1 FINE-TUNING *DARE* IN THE VOLCANO MODEL.

We embed *DARE* into the 7B-parameter VolCano model (Li et al., 2025b), which interleaves textual tokens with RefBind–based visual tokens for fine-grained grounding. Instruction tuning is performed on the *VoCoT* corpus under the configuration summarized in Table 9. Briefly, we keep the **CLIP ViT-L/14** visual encoder frozen (input resolution $336 \times 336$) and initialize the language backbone with multi-domain caption checkpoints (ALLaVA-Caption, GRIT, Flickr30k-Entities, MMC4). Training uses the *AdamW* optimiser (Loshchilov and Hutter, 2017) with $\beta_1 = 0.9$, $\beta_2 = 0.95$, $\epsilon = 10^{-4}$, a base learning rate of $1 \times 10^{-4}$ for visual-router parameters (and a peak LLM LR of $1 \times 10^{-5}$), weight decay $3 \times 10^{-2}$, no warm-up, and a cosine schedule. A global batch of 128 sequences (each $\leq 3072$ tokens) is optimised for one epoch, using **bfloat16** precision on NVIDIA A100 GPUs; gradients are clipped

Table 9: *DARE*'s training configuration for the instruction tuning stage.

| Configuration | Instruction Tuning |
|---|---|
| Visual Encoder | OpenAI-CLIP ViT-L/14 (Radford et al., 2021) |
| Backbone Init | ALLaVA-Caption (Chen et al., 2024a), GRIT (Peng et al., 2023) |
| | Flickr30k-Entities (Plummer et al., 2015), MMC4 (Zhu et al., 2023) |
| Optimizer | *AdamW* |
| Optimizer Hyperparameters | $\beta_1 = 0.9,\ \beta_2 = 0.95,\ \epsilon = 1\mathrm{e}{-4}$ |
| Global batch size | 64 |
| Peak learning rate of LLM | 1e-5 |
| Learning rate schedule | Cosine |
| Training Epochs | 1 |
| Warm-up ratio | 0 |
| Weight decay | $3 \times 10^{-2}$ |
| Gradient clipping | 1.0 |
| Input image resolution | $336 \times 336$ |
| Input sequence to LLM | 3072 |
| Numerical precision | bfloat16 |
| GPU Usage | $4 \times 4$ NVIDIA A100s |
| Training Time | 115h |

to 1.0, and total wall-clock time is $\sim$30 h. Under this regime, *DARE* jointly minimizes task loss plus sparsity regularizers, converging to retention ratios of roughly $40\%$ for visual tokens and $70\%$ for text tokens, while preserving VolCano's task accuracy and reducing both FLOPs and KV cache by $\geq 40\%$.

After completing instruction tuning, we evaluate the fine-tuned *DARE*-VolCano model on a suite of downstream benchmarks covering static spatial reasoning, dynamic spatial reasoning, general visual question answering, and hallucination detection. For spatial reasoning, we assess performance on VSR, EmbSpatial, V-Star, and Winoground. For general VQA, we use GQA with the "testde_balanced" split to measure compositional question answering performance. Finally, we assess factual grounding and hallucination resistance using POPE (binary classification) and AMBER (generative description), which stress the model's ability to align outputs with visual evidence.

## C.2 Fine-tuning *DARE* on Anole-7B Model

We integrate *DARE* into the Anole-7B model (Chern et al., 2024), a multimodal architecture that combines textual inputs with self-generated mental imagery to support abstract spatial reasoning. We fine-tune this model for 60 epochs on three challenging multi-hop, dynamic spatial reasoning tasks: MAZE (Ivanitskiy et al., 2023), MINIBEHAVIOR (Jin et al., 2023), and FROZENLAKE (Wu et al., 2024a). These tasks require the model to reason over evolving spatial environments and multi-step action trajectories. Fine-tuning follows the MVoT configuration shown in Table 10, using *AdamW* with $\beta_1$=0.9, $\beta_2$=0.95, and $\epsilon$=2e$-$4, weight decay $3\times10^{-2}$, and a batch size of 8 with gradient accumulation of 2. Training is distributed across 4 NVIDIA GPUs.

This setup enables *DARE* to dynamically prune uninformative tokens while preserving crucial trajectory and spatial cues. By integrating *DARE*'s retention-aware routing into Anole-7B, we demonstrate that our method generalizes beyond static instruction-tuned models, enabling scalable and efficient reasoning in long-horizon, procedurally generated environments.

## C.3 Implementation of Baselines

We adopt three recent efficiency–oriented reasoning baselines, *SoT*, *LightFast*, and *Heima*, and align their settings with our *DARE* evaluation pipeline.

**SoT (Sketch-of-Thought).** SoT is a static prompt framework that condenses reasoning into cognitively inspired "sketch" phrases, reducing token usage by a predefined retention ratio $\rho$ without model fine–tuning (Aytes et al., 2025). For each task, we prepend the task–specific SoT instruction ("`<sketch>`") to VolCano's original prompt, explicitly injecting the same retention ratio $\rho$ used in *DARE* to ensure consistent sparsity constraints. We maintain the same generation parameters (temperature 0.7). Because SoT is purely prompt-based, no additional training or model modification is required.

**LightFast (LightThinker (Zhang et al., 2025a) + FastV) (Chen et al., 2024c).** LightFast combines *LightThinker*, a text-side token compression module that prunes low-utility reasoning tokens, with *FastV*, a visual pruning component that discards redundant visual tokens after the second transformer layer. We implement LightThinker as a lightweight controller atop VolCano's language blocks, using the original hyperparameters (retention ratio matched to *DARE*, $\lambda_{\text{entropy}} = 5\text{e}^{-3}$). FastV is integrated into the frozen CLIP encoder with its default layer-2 cutoff strategy. Both modules are fine-tuned jointly for one epoch on the VoCoT corpus to ensure a fair comparison with *DARE* under the same training budget. This setup reflects a strong modular baseline that simulates independent, modality-specific compression without unified routing.

**Heima.** Heima shifts chain-of-thought reasoning into a latent "hidden thinking" space by compressing intermediate reasoning into a single *thinking token*, which is decoded only at the final step (Shen et al., 2025). We condition the Heima Encoder on interleaved image and text inputs, allowing the latent reasoning vector to capture fused cross-modal semantics to adapt it for multimodal reasoning. Each intermediate reasoning step is encoded as a 128-dimensional vector, while a frozen Heima Decoder reconstructs the final output if needed. We follow the official implementation, fine-tuning only the encoder using *AdamW* ($\beta_1$=0.9, $\beta_2$=0.95, LR $1\times10^{-4}$) for 3 epochs on the VoCoT corpus. This setup offers a strong latent-space baseline that eliminates intermediate tokens while leveraging multimodal context.

**SparseVLM.** *SparseVLM* introduces a visual token sparsification framework that prunes redundant image tokens before they are fed into the language backbone (Zhang et al., 2025c). We integrate SparseVLM into the VolCano pipeline by applying its token selection module on CLIP-encoded visual features, using the same retention ratio $\rho$ as in *DARE* to ensure fairness. This module ranks tokens by learned importance scores and discards background or low-saliency patches while preserving critical object-centric cues. Following the official setup, the pruning module is jointly fine-tuned with VolCano for one epoch on the VoCoT corpus under identical optimization settings (AdamW, $\beta_1$=0.9, $\beta_2$=0.95, LR $1\times10^{-4}$). This configuration reflects a strong visual sparsification baseline that reduces computation and KV-cache usage without altering the textual pathway.

Table 10: Hyperparameters used for fine-tuning Anole 7B.

| Configurations | *DARE* |
|---|---|
| Random Seed | 42 |
| Epochs | 60 |
| Optimizer | *AdamW* |
| Optimizer Hyperparameters | $\beta_1 = 0.9,\ \beta_2 = 0.95,\ \epsilon = 2\text{e}{-}4$ |
| Weight decay | $3 \times 10^{-2}$ |
| Train Batch Size | 8 |
| Val Batch Size | 8 |
| Grad Accumulation | 2 |

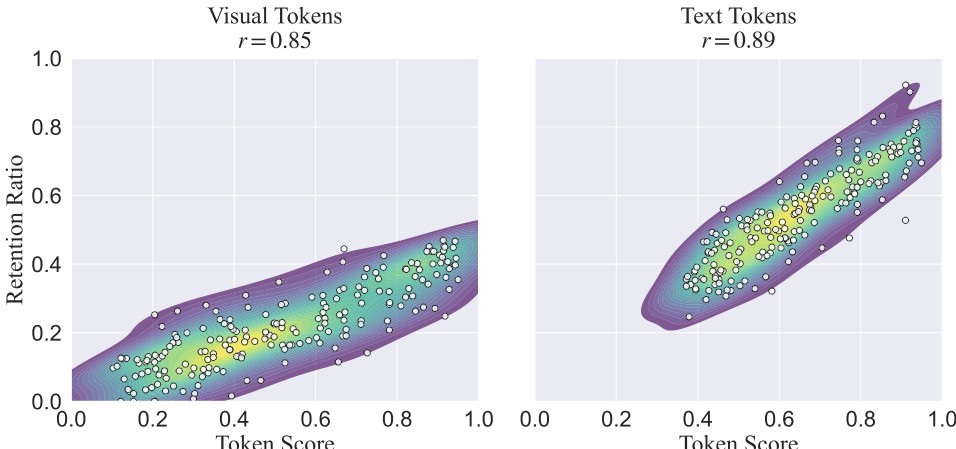

Figure 6: Scatter-density plots of router-predicted token scores vs. learned retention ratios for *visual* and *text* tokens. Pearson correlations ($r = 0.85$ visual; $r = 0.89$ text) indicate that *DARE* aligns token importance with retention, despite different sparsity budgets (40 % visual, 75 % text).

## D    MECHANISMS AND ANALYSIS OF LEARNABLE TOKEN RETENTION

### D.1    CORRELATION BETWEEN LEARNED RETENTION RATIOS AND ROUTER PREDICTIONS

Figure 6 illustrates the relationship between token importance scores predicted by the *DARE* router and the corresponding retention ratios after differentiable token compression. Each point represents a token, with its router score on the x-axis and its final retention ratio on the y-axis. Pearson correlation coefficients are computed over $\sim$25,000 tokens spanning 5 hops and 32 layers on the validation set, while 200 tokens in each subfigure are uniformly sampled for visualization in the figure. We observe strong positive correlations ($r = 0.85$ for visual tokens, $r = 0.89$ for text tokens), despite differing average retention budgets (40% for visual tokens and 75% for text tokens). This trend confirms that higher router scores consistently translate into higher retention probabilities. Importantly, this alignment emerges without any explicit supervision, validating that *DARE*'s joint routing-compression framework effectively couples token scoring with sparsity-aware retention across heterogeneous modalities.

### D.2    EFFECTIVENESS OF MSE LOSS FOR RETENTION PREDICTION

*DARE* relies on a lightweight *mean-squared-error* (MSE) objective to steer its router toward a desired sparsity profile. For each modality $m \in \{t, v\}$, hop $h$, and layer $l$, the auxiliary loss $\mathcal{L}_{\text{ratio}}^{(m)} = (\hat{\rho}_{h,l}^{(m)} - \rho_{\text{target}}^{(m)})^2$ penalizes deviations between the *empirical* retention ratio $\hat{\rho}_{h,l}^{(m)}$ and a *target* ratio $\rho_{\text{target}}^{(m)}$. Taking the gradient with respect to the learnable ratio parameter yields

$$\frac{\partial \mathcal{L}_{\text{ratio}}^{(m)}}{\partial \rho_{h,l}^{(m)}} = 2\big(\hat{\rho}_{h,l}^{(m)} - \rho_{\text{target}}^{(m)}\big), \tag{11}$$

which is an unbiased estimator of the error magnitude. Hence the update magnitude *shrinks linearly* as the observed ratio approaches the target, guaranteeing a first-order stationary point at $\hat{\rho} = \rho_{\text{target}}$. Because $\mathcal{L}_{\text{ratio}}^{(m)}$ is convex in $\rho$, gradient descent with a diminishing step size converges to this optimum under the same Lipschitz conditions stated in Proposition 1 (Sec. G). In practice we observe rapid stabilization of the per-layer retention ratios ($\leq 5$ training epochs), providing a reliable sparsity signal to the router while adding negligible computational overhead.

Table 11: **Ablation on modality-specific MSE losses.** Accuracy (%, ↑) after (i) removing the text-ratio loss, (ii) removing the visual soft + hard losses, and (iii) removing *all* MSE regularisers.

| Benchmark | All MSE | $-\mathcal{L}_{\text{ratio}}^{(t)}$ | $-(L_{\text{soft}}^{(v)}+L_{\text{hard}}^{(v)})$ | $-$Both |
|---|---|---|---|---|
| EmbSpatial | **68.09** | 67.12 | 64.42 | 60.01 |
| Winoground | **68.31** | 67.01 | 61.78 | 57.34 |
| V-Star | **60.07** | 56.21 | 52.52 | 49.21 |
| VSR | **68.09** | 66.62 | 62.31 | 59.23 |
| MAZE | **93.32** | 90.90 | 82.73 | 76.42 |
| MINIBEHAVIOR | **95.47** | 92.11 | 85.02 | 70.71 |
| FROZENLAKE | **86.11** | 83.13 | 77.61 | 69.21 |
| **Avg. $\Delta$** | — | $-2.34$ | $-7.58$ | $-13.90$ |

**Empirical Evidence.** Table 11 presents an ablation study on the role of modality-specific MSE regularizers in *DARE*. Removing the text-token ratio loss $\mathcal{L}_{\text{ratio}}^{(t)}$ results in a 2.34% average accuracy drop, with notable degradation on V-Star and Winoground. Excluding the visual retention losses $\mathcal{L}_{\text{soft}}^{(v)} + \mathcal{L}_{\text{hard}}^{(v)}$ causes a larger 7.58% drop, underscoring their role in preserving spatial precision. Removing all MSE terms leads to a 13.90% average decline, confirming that task supervision alone is insufficient for effective sparsity-aware routing.

These results show that the MSE regularizers play a critical role in guiding *DARE*'s router toward consistent and effective token selection. Without these losses, the model tends to misalign token importance scores with retention behavior, leading to degraded accuracy across both visual and textual modalities. As illustrated in Figure 6, the inclusion of MSE supervision enhances the correlation between router scores and final retention decisions. This complements our theoretical analysis, where the convex form of the MSE objective ensures stable optimization toward the target sparsity. Overall, the empirical and theoretical evidence supports the inclusion of modality-specific MSE losses as essential components for reliable and efficient routing.

## D.3 WHY GUMBEL-SOFTMAX INSTEAD OF SOFTMAX?

**Why Gumbel-Softmax?** We employ Gumbel-Softmax (Jang et al., 2017) to enable differentiable yet sparse token selection, which standard softmax cannot provide. Unlike softmax, which always outputs dense, positive weights that retain all tokens and thus prevents any FLOP or KV-cache reduction (Table 12), Gumbel-Softmax offers three advantages: (1) inference-time sparsity, where setting $\tau = 0$ yields deterministic top-$k$ token selection and allows token skipping in attention/MLP layers, (2) stable gradients, as the added i.i.d. Gumbel(0,1) noise and temperature $\tau$ produce a smooth, differentiable relaxation unlike non-differentiable softmax+thresholding, and (3) improved exploration early in training, since the injected noise prevents premature collapse on arbitrarily ranked tokens.

Table 12: Comparison between Softmax and Gumbel-Softmax.

| Operation | Forward-pass output | Gradients w.r.t. logits |
|---|---|---|
| Softmax | Dense, all-positive weights that sum to 1. | Non-zero, but every token is still processed; no FLOP or KV-cache savings. |
| Gumbel-Softmax (add Gumbel noise, divide by $\tau$, softmax) | For $\tau \approx 1$, behaves like a noisy softmax (encouraging exploration); as $\tau \to 0$, converges to one-hot vectors, mimicking top-$k$. | Non-zero until $\tau \to 0$, enabling end-to-end training of the router and retention ratios $\rho$. |

**Empirical evidence.** Our ablation study (Table 13) highlights the necessity of Gumbel-Softmax. Replacing it with softmax+thresholding reduces average accuracy by over 6% (77.01% → 70.92%) without any gain in KV-cache savings, while fixing $\tau = 1$ throughout leads to even larger performance drops and weaker memory reduction (–34% vs. –46%). These findings demonstrate that Gumbel-Softmax is crucial for maintaining both stable training dynamics and efficient inference, enabling deterministic top-$k$ token selection in *DARE*.

Table 13: Ablation on Gumbel-Softmax. Accuracy is averaged across 9 tasks.

| Variant | Avg. Accuracy (9 tasks) | KV-cache $\Delta$ |
|---|---|---|
| Full *DARE* (Gumbel-Softmax) | **77.01%** | **–46%** |
| Replace with softmax+threshold | 70.92% | –37% |
| No perturbation, $\tau = 1$ throughout | 70.73% | –34% |

### D.4 COMPARISON WITH HEURISTIC TOKEN RETENTION AND ROUTING METHODS

We benchmarked *DARE* against four budget-matched heuristic token retention strategies to evaluate the effectiveness of learned routing: (1) Gist token (Mu et al., 2023), which compresses input sequences into a summary token, (2) average attention scores, which retain tokens with the highest mean attention across heads, (3) hidden-state norm pruning, which removes tokens with low activation magnitudes, and (4) heavy-hitter routing (Zhang et al., 2023), which prioritizes tokens frequently activated across layers.

Table 14: Comparison of heuristic routers and *DARE* on VSR and MAZE benchmarks.

| Router Type | VSR Acc. | MAZE Acc. | Avg. GFLOPs $\times 10^3$ |
|---|---|---|---|
| Gist token | 62.8% | 84.5% | 14.8 |
| Hidden-state norm | 57.2% | 83.6% | 14.3 |
| Mean attention score | 60.7% | 81.8% | 15.5 |
| Heavy-hitter | 63.0% | 88.7% | 13.2 |
| *DARE* (learned) | **68.1%** | **93.3%** | **10.1** |

As shown in Table 14, heuristic methods underperform *DARE* by 4.6–11.5% in accuracy, while also incurring higher computational cost per token retained. The primary limitation of these heuristics is that they operate in a layer-wise manner, without adapting to evolving cross-hop or cross-modal dependencies, and are unable to dynamically detect task-dependent information cliffs. In contrast, *DARE*'s learned routing mechanism provides consistent gains in both efficiency and accuracy across benchmarks.

We further compared *DARE* with the heuristic MoD (Raposo et al., 2024) routing mechanism. MoD prunes tokens using fixed, hard-coded top-$k$ ratios at each layer, designed primarily for unimodal, single-pass inference. This approach lacks the ability to adapt retention across layers, reasoning hops, or modalities, and does not address memory growth from recurrent reasoning.

By contrast, *DARE* represents a principled departure from heuristic-based MoD and introduces four key innovations. First, it implements **modality-aware routing** through asymmetric routing heads for vision and text. This enables pruning decisions to adapt to each modality's changing informativeness after fusion, whereas MoD is restricted to unimodal routing and cannot distinguish modality-specific contributions. As a result, *DARE*-LH outperforms the unimodal UniPru baseline by up to +9.1% accuracy while simultaneously reducing FLOPs by as much as 32% (Table 15).

Second, *DARE* employs **dynamic, intra- and inter-hop-aware retention ratio**. Its retention ratios are learnable and differentiated by both hop and layer, optimized end-to-end via Gumbel-Softmax. In contrast, MoD applies fixed, hard-coded top-$k$ ratios per layer, preventing adaptation to evolving reasoning dynamics. In matched-budget comparisons, *DARE* improves accuracy by up to +8.7% over MoD (EmbSpatial), as shown in Table 16. Further evidence in Table 17 shows that hop-awareness alone provides an additional +4.2% accuracy and –21.6% FLOPs savings compared to *DARE*-L, underscoring the importance of intra- and inter-aware retention rates.

Third, *DARE* explicitly aligns routing with the **information fusion process across modalities**. Multi-hop spatial reasoning requires joint processing of vision and text, where visual cues are progressively absorbed into the textual stream. *DARE* captures this transition: early layers retain more visual tokens for grounding and spatial alignment, while deeper layers prioritize text tokens for abstract reasoning.

Finally, *DARE* addresses the often-overlooked challenge of **KV-cache efficiency in recurrent reasoning**. In multi-hop settings, recurrent computation causes rapid growth in KV-cache size, which in practice, rather than FLOPs, is often the dominant bottleneck for scalability. MoD entirely overlooks this issue. *DARE* mitigates it with an execution mask (Eq. 9) that prunes dropped tokens from the cache at every hop, reducing memory requirements by 35–48% (Table 18). This mechanism substantially improves scalability and makes *DARE* well-suited for long-horizon reasoning tasks where cache size, not compute, is the limiting factor.

Table 15: Comparison of *DARE*-LH with unimodal UniPru baseline.

| Task | $\Delta$ Acc. (*DARE*-LH $-$ UniPru) | *DARE*-LH FLOPs Saving |
|---|---|---|
| VSR | +4.38 | −20.6% |
| V-Star | +6.85 | −18.3% |
| EmbSpatial | +7.34 | −15.7% |
| Winoground | +2.48 | −14.1% |
| MAZE | +9.10 | −24.9% |
| MiniBehavior | +5.13 | −24.4% |
| FrozenLake | +2.00 | −32.0% |

Table 16: Comparison between MoD and *DARE* under matched budgets.

| Task | MoD | *DARE* | $\Delta$ Acc. (*DARE*$-$MoD) |
|---|---|---|---|
| VSR | 62.11 | 68.09 | +5.98 |
| V-Star | 50.72 | 60.07 | +9.35 |
| EmbSpatial | 59.37 | 68.09 | +8.72 |
| Winoground | 62.65 | 68.31 | +5.66 |

Table 17: Accuracy and FLOP savings of *DARE*-LH relative to *DARE*-L.

| Task | $\Delta$ Acc. (*DARE*-LH $-$ *DARE*-L) | *DARE*-LH FLOPs Saving |
|---|---|---|
| VSR | +0.96 | −11.8% |
| V-Star | +2.68 | −10.2% |
| EmbSpatial | +3.72 | −8.5% |
| Winoground | +1.09 | −7.8% |
| MAZE | +3.54 | −20.2% |
| MiniBehavior | +4.22 | −20.4% |
| FrozenLake | +2.00 | −21.6% |

These results confirm that *DARE* takes a **principled departure from heuristic-based MoD**. Its learnable and modality-aware design, aligned with cross-modal fusion and adaptive across hops, not only delivers substantial accuracy gains but also addresses KV-cache efficiency, a limitation unaddressed by MoD. This distinction is crucial for efficient multi-hop multimodal reasoning and explains *DARE*'s consistent superiority across benchmarks, as heuristic approaches tend to accumulate and propagate errors while lacking the fine-grained signals required to calibrate compression at the level of each layer, hop, and modality.

### D.5 STABILITY OF LEARNED RETENTION MASKS ACROSS SEEDS

We trained the model five times with independent random seeds (11111, 22222, 33333, 44444, 55555) to evaluate the robustness of *DARE* to initialization.

Table 18: KV-cache memory usage (GB) across hops in VSR and related tasks.

| Task | Thought 1 | | Thought 3 | | Thought 5 | |
|---|---|---|---|---|---|---|
| | MoD | *DARE* | MoD | *DARE* | MoD | *DARE* |
| VSR | 1.92 | 1.21 | 3.28 | 2.09 | 5.69 | 2.97 |
| V-Star | 1.96 | 1.35 | 3.34 | 2.37 | 5.52 | 3.28 |
| EmbSpatial | 2.01 | 1.51 | 3.41 | 2.65 | 5.37 | 3.52 |
| Winoground | 2.05 | 1.62 | 3.48 | 2.86 | 5.72 | 3.60 |

**Accuracy stability.**    As shown in Table 19, performance remains highly consistent across all seven spatial reasoning tasks. For instance, VSR achieves $68.3 \pm 0.24$, and all other tasks exhibit standard deviations below 0.32 points, which is well within expected random variation. These results confirm that *DARE* maintains stable accuracy regardless of initialization.

**Retention mask consistency.**    In addition to accuracy, the learned retention masks are also reproducible across seeds. Specifically, the average Jaccard overlap of retained tokens across seed pairs is $0.91 \pm 0.02$, while the Spearman correlation between router scores exceeds $\rho = 0.94$ across all layers and modalities. This demonstrates that *DARE* consistently learns near-identical retention patterns across runs, further reinforcing its stability and reliability.

Table 19: Accuracy stability of *DARE* across random seeds. Reported as mean $\pm$ standard deviation.

| Metric (across seeds) | VSR | V-Star | EmbSpatial | Winoground |
|---|---|---|---|---|
| Mean $\pm\sigma$ | $68.3 \pm 0.24$ | $60.4 \pm 0.06$ | $68.1 \pm 0.07$ | $68.4 \pm 0.17$ |
| **Metric (across seeds)** | **MAZE** | **MiniBehavior** | **FrozenLake** | |
| Mean $\pm\sigma$ | $93.5 \pm 0.09$ | $95.8 \pm 0.11$ | $86.4 \pm 0.32$ | |

## D.6    DETAILS OF RETENTION RATIOS $\rho_{h,l}^{(m)}$, TARGETS $\rho_{\text{TARGET}}^{(m)}$, AND THEIR RATIONALE

We provide additional clarification on the retention ratios used in *DARE*. The framework distinguishes between three quantities: (i) learnable per-hop, per-layer, per-modality ratios $\rho_{h,l}^{(m)}$, (ii) the corresponding empirical ratios $\hat{\rho}_{h,l}^{(m)}$ realized during execution, and (iii) modality-specific global targets $\rho_{\text{target}}^{(m)}$ that act as sparsity anchors.

**Learnable vs. empirical ratios.**    Each ratio $\rho_{h,l}^{(m)}$ is a **learnable scalar parameter**, defined separately for each modality $m \in \{\text{text, vision}\}$, hop $h$, and layer $l$. These parameters are initialized to zero and updated jointly with the model via backpropagation. During the forward pass, the router produces token importance scores $s_{h,l}^{(i,m)}$ (Eq. 2). A differentiable Gumbel–Softmax mask (Eq. 4) selects the top fraction specified by $\rho_{\text{target}}^{(m)}$, producing routed activations $\tilde{y}_{h,l}^{(i,m)}$ (Eq. 3). The **empirical retention ratio** is then

$$\hat{\rho}_{h,l}^{(m)} = \frac{1}{N_{h,l}^{(m)}} \sum_{i=1}^{N_{h,l}^{(m)}} \alpha_{h,l}^{(i,m)}, \tag{12}$$

where $N_{h,l}^{(m)}$ is the token count and $\alpha_{h,l}^{(i,m)} \in \{0, 1\}$ is the selection mask. Thus, $\rho_{\text{target}}^{(m)}$ encodes the intended budget, while $\hat{\rho}_{h,l}^{(m)}$ reflects the actual fraction realized in the forward pass.

**Consistency with global targets $\rho_{\text{target}}^{(m)}$.**    We introduce modality-specific global anchors $\rho_{\text{target}}^{(m)}$ that regulate average retention across hops and layers. Without these anchors, the learned ratios $\rho_{h,l}^{(m)}$ drift upward during fine-tuning, leading to excessive memory use (up to $2.7\times$ KV-cache growth on *VSR* and nearly $2\times$ on average across seven multimodal spatial reasoning tasks).

A quadratic auxiliary loss

$$\mathcal{L}_{\text{budget}}^{(m)} = \frac{1}{HL} \sum_{h=1}^{H} \sum_{l=1}^{L} \left( \hat{\rho}_{h,l}^{(m)} - \rho_{\text{target}}^{(m)} \right)^2, \tag{13}$$

softly penalizes deviations from the global budget while preserving local flexibility. These anchors stabilize FLOP and memory usage, prevent error accumulation in autoregressive reasoning, and keep KV-cache growth under control. With $\rho_{\text{target}}^{(t)} = 0.7$ and $\rho_{\text{target}}^{(v)} = 0.40$, *DARE* achieves large efficiency gains ($\geq 40\%$ FLOP reduction and $\geq 46\%$ KV-cache savings) without sacrificing accuracy.

**Sensitivity analysis.** We validated robustness by sweeping text targets $\rho_{\text{target}}^{(t)}$ from 60–80% and visual targets from 20–60% (Figure 5). Accuracy remained within $1\%$ of baseline across most settings, degrading only when the text ratio fell below $20\%$. The 70%/40% configuration provides the best trade-off between accuracy and efficiency. Under these targets, *DARE* consistently outperforms or matches baselines on nine benchmarks while maintaining over 40% FLOP reduction and 46% KV-cache savings.

## D.7 DATA-DRIVEN DETERMINATION AND ROBUSTNESS OF THE VISUAL PRUNING PHASE $l_c$

*DARE*'s transition from soft to hard visual token pruning is **not heuristic**, but an automated, data-driven procedure that generalizes across models and tasks. The method is described in Alg. 4.

**Automated detection.** The pruning phase boundary $l_c$ is determined at the start of training by analyzing the router's layer-wise mean visual token importance. Specifically, the importance curve is first smoothed with a short moving average. A greedy scan is then applied to identify the first layer where importance consistently falls below a fixed threshold $\epsilon$, after which hard pruning is applied. This process is transparent, deterministic, and requires no manual tuning. As a result, $l_c$ naturally adapts across architectures and scales (e.g., $l_c = 14$ for VSR-7B, $l_c = 18$ for Chameleon-34B).

**Robustness of $l_c$.** The boundary is highly robust to hyperparameters and offsets. Shifting $l_c$ by $\pm 3$ layers alters accuracy by at most $0.3\%$ and FLOP savings by at most $1\%$, confirming that $l_c$ serves as a stable diagnostic rather than a sensitive knob (Table 20). Across ten diverse benchmarks, the detected thresholds $\epsilon$ are consistently below $0.01$, and $l_c$ typically lies near the midpoint of the network (Table 21), reflecting consistent dynamics in visual-token utility.

**Comparison with learned gates.** We also experimented with learning $l_c$ using a differentiable gating mechanism. While the gate converged to a similar location (within $\pm 2$ layers of the automatically detected $l_c$), it introduced gradient noise and increased variance ($\pm 0.3\%$ accuracy fluctuations) without providing any performance gains. This demonstrates that the automated procedure is not only simpler, but also more reliable.

**Interpretation.** The automatically detected split aligns with a genuine semantic transition in multimodal reasoning. As shown in Figure 3, visual token importance drops sharply after $l_c$, exactly when text importance rises, marking the point where visual features have been absorbed and linguistic reasoning becomes dominant. Thus, *DARE*'s pruning boundary reflects the natural dynamics of information flow, ensuring that pruning decisions remain both effective and semantically grounded.

Table 20: Robustness of *DARE* to shifts in the pruning boundary $l_c$. Accuracy (%) and FLOP savings are reported.

| Offset from $l_c$ | VSR Acc. | FLOPs↓ | MAZE Acc. | FLOPs↓ |
|---|---|---|---|---|
| −3 | 68.0 | −40.8% | 93.2 | −41.5% |
| 0 (auto) | **68.1** | **−41.6%** | **93.3** | **−42.1%** |
| +3 | 67.9 | −40.9% | 93.1 | −41.3% |

Table 21: Post-jump importance values and corresponding $l_c$ values across benchmarks.

| Benchmark | Post-jump importance values | $l_c$ (layer index) |
|---|---|---|
| VSR | 0.003 | 15 |
| V-Star | 0.007 | 14 |
| EmbSpatial | 0.002 | 15 |
| Winoground | 0.003 | 15 |
| MAZE | 0.004 | 14 |
| MiniBehavior | 0.003 | 13 |
| FrozenLake | 0.007 | 13 |
| NLVR2 | 0.010 | 15 |
| GQA | 0.009 | 15 |
| POPE | 0.002 | 14 |

# E    EFFICIENCY ANALYSIS

## E.1    PARAMETER OVERHEAD

*DARE* introduces only two categories of additional parameters: (i) lightweight per-layer routing heads, and (ii) learnable retention ratios $\rho$ that determine the fraction of tokens preserved per modality, hop, and layer. For a 7B-parameter VolCano backbone with $L = 32$ layers, hidden size $d = 4096$, $H = 5$ reasoning hops, and $M = 2$ modalities, the total number of added parameters is

$$P_{DARE} = L \times M \times (d + 1) + H \times L \times M = 262{,}528.$$

This accounts for less than $0.004\%$ of the base model size, over three orders of magnitude smaller than typical LoRA adapters. Such an overhead is negligible, confirming that *DARE*'s efficiency gains are achieved without materially increasing model size or training complexity.

## E.2    TRAINING STABILITY AND CONVERGENCE SPEED

*DARE* maintains stable optimization while accelerating training efficiency. As shown in Table 22, training across five independent random seeds exhibited no instability events, with accuracy variance below $0.70\%$. Moreover, *DARE* reduced wall-clock time to the same validation loss by $32\%$, primarily due to early token pruning, which decreases per-layer computation by $25$–$40\%$ from the first mini-batch onward. Importantly, convergence speed in terms of optimization steps remains nearly unchanged (17.5k vs. 17.2k), indicating that faster training arises from lower per-step cost rather than shallower optimization.

Table 22: Training metrics for *DARE* vs. baseline Anole-7B.

| Aspect | Baseline Anole-7B | *DARE*-Anole-7B |
|---|---|---|
| Extra parameters | — | +0.26M |
| Per-step FLOPs (train) | 100% | **57–69%** |
| Wall-clock time to same val. loss | 28h | **19h (–32%)** |
| Convergence steps (to 99% final acc.) | 17.2k | 17.5k |
| Instability events (loss spikes $>5\times$ median, 5 seeds) | 0/5 | 0/5 |

**Sources of stability.**    The robustness of *DARE* is attributable to three simple design factors. First, the auxiliary retention-target losses are deliberately down-weighted ($0.1\times$) and removed after the early epochs; excluding them changes final accuracy by less than $0.3\%$, confirming their role as gentle regularizers. Second, the Gumbel-Softmax relaxation with moderate temperature ($\tau = 0.7$) provides smooth gradients and stable training; performance remains consistent across $\tau \in [0.5, 0.9]$, while hard masking is applied only at inference. Finally, *DARE* reuses the same optimizer (AdamW) and schedule as the baseline model, requiring no additional warm-up phases or learning rate groups. These choices ensure that *DARE* integrates seamlessly into standard training pipelines, achieving significant efficiency gains without sacrificing stability or convergence quality.

### E.3 Engineering Effort

*DARE* integrates with existing MLLM frameworks with minimal overhead. The implementation introduces a lightweight router module responsible for three tasks: (i) token importance scoring, (ii) differentiable soft masking during training, and (iii) KV-cache pruning at inference. All components are implemented at the Python layer, requiring no modifications to attention kernels, checkpoint formats, or training loops. Consequently, *DARE* can be incorporated as a plug-and-play option with only a few hundred lines of additional code, leaving the backbone architecture and training pipeline entirely unchanged.

## F Hyperparameter Sensitivity and Ablation

### F.1 Ablation on pruning threshold $\epsilon$

We further evaluated the sensitivity of *DARE* to the pruning threshold $\epsilon$ on both a static benchmark (*VSR*) and a dynamic benchmark (*MAZE*). As shown in Table 23, varying $\epsilon$ over a wide range produces negligible changes in performance: accuracy shifts by less than $0.2\%$ and FLOP savings by less than $1\%$. This demonstrates that *DARE*'s efficiency and accuracy are highly robust to the choice of $\epsilon$, making the method stable across different settings without the need for hyperparameter tuning.

Table 23: Sensitivity of *DARE* to $\epsilon$ on VSR and MAZE. Accuracy (%) and FLOP savings are reported.

| $\epsilon$ | VSR Acc. | $\Delta$ | FLOPs↓ | MAZE Acc. | $\Delta$ | FLOPs↓ |
|---|---|---|---|---|---|---|
| 0 (no penalty) | 67.9 | −0.2 | −40.1% | 93.1 | −0.1 | −41.2% |
| 0.005 | **68.1** | 0 | **−41.6%** | 93.1 | −0.2 | −42.0% |
| 0.010 (def.) | **68.1** | 0 | **−41.6%** | **93.3** | 0 | **−42.1%** |
| 0.020 | 68.0 | −0.1 | −41.3% | 93.2 | −0.1 | −41.9% |
| 0.050 | 67.7 | −0.4 | −40.8% | 93.1 | −0.2 | −41.0% |

### F.2 Impact of Temperature $\tau$ Scaling

Temperature $\tau$ in *DARE* controls the sharpness of the token importance distribution produced by the Gumbel-Softmax routing mechanism. Lower temperatures lead to harder, more discrete selections, while higher temperatures introduce softer token scores. Table 24 presents an ablation over $\tau \in \{0.3, 0.5, 0.7, 0.9, 1.1, 1.3\}$ across all seven benchmarks. We observe that both excessively low and high temperatures degrade performance slightly, likely due to increased noise or over-smoothing in the token selection process.

A moderate temperature of $\tau = 0.7$ consistently yields the best accuracy across most datasets, balancing sharpness and stability in routing decisions. Notably, this value performs well across both static (e.g., EmbSpatial, VSR) and dynamic spatial reasoning tasks (e.g., MAZE, FROZENLAKE), indicating that the routing mechanism generalizes effectively under shared hyperparameter settings. These results validate our choice of a fixed temperature during training and confirm that *DARE*'s differentiable routing is robust to moderate variations in $\tau$.

## G Theoretical Analysis

We provide a formal analysis of the retention mechanism in *DARE*, showing (i) that the hop–layer–modality ratios $\rho_{h,l}^{(m)}$ optimized with the *AdamW* algorithm converge, asymptotically driving the expected gradient norm $|\nabla_\rho \mathcal{L}|$ to zero, and (ii) that the Gumbel–Softmax relaxation employed during training approximates the hard top-$k$ masking used at inference within an $\mathcal{O}(\tau)$ gap, where $\tau$ is the final temperature.

Table 24: **Impact of routing temperature $\tau$ on accuracy.** Lower or higher $\tau$ values lead to slightly noisier routing decisions, while $\tau = 0.7$ provides the most balanced and robust results across tasks.

| Temperature $\tau$ | EmbSpatial$^\uparrow$ | Winoground$^\uparrow$ | V-Star$^\uparrow$ | VSR$^\uparrow$ | MAZE$^\uparrow$ | MINIBEHAVIOR$^\uparrow$ | FROZENLAKE$^\uparrow$ |
|---|---|---|---|---|---|---|---|
| 0.3 | 67.81 | 67.42 | 59.33 | 67.58 | 92.97 | 94.95 | 85.62 |
| 0.5 | 68.02 | 67.91 | 59.75 | 67.92 | 93.21 | 95.12 | 85.79 |
| 0.7 | **68.09** | **68.31** | **60.07** | **68.07** | **93.32** | 95.47 | **86.11** |
| 0.9 | 67.95 | 68.14 | 59.82 | 67.89 | 93.17 | **95.47** | 85.97 |
| 1.1 | 67.63 | 67.55 | 59.01 | 67.41 | 92.84 | 94.87 | 85.44 |
| 1.3 | 67.29 | 67.10 | 58.76 | 66.98 | 92.53 | 94.52 | 85.03 |

### G.1 NOTATION AND SETUP

Let $\Theta$ denote the model parameters excluding the retention variables $\rho$. The full loss is decomposed as:

$$\mathcal{L}(\Theta, \rho) = \mathcal{L}_{\text{task}} + \mathcal{L}_{\text{ratio}}^{(t)} + \mathcal{L}_{\text{soft}}^{(v)} + \mathcal{L}_{\text{hard}}^{(v)}. \tag{14}$$

Each auxiliary term regularizes the learned retention schedule $\rho$ to promote sparsity in a structured and adaptive way. For hop $h$, layer $l$, and modality $m \in \{t, v\}$, let $s_{h,l}^{(i,m)}$ be the router score for token $i$, $\alpha_{h,l}^{(i,m)} \in \{0, 1\}$ the top-$k$ binary mask at inference, and $\tilde{\alpha}_{h,l}^{(i,m)} \in (0, 1)$ the soft score during training.

**Assumptions:**

**A1 (*Lipschitz Continuity*).** The gradient $\nabla_\rho \mathcal{L}$ is $L$-Lipschitz continuous.

**A2 (*Bounded Variance*).** There exists $G > 0$ such that $\mathbb{E}[\|\nabla_\rho \mathcal{L}\|^2] \leq G^2$.

### G.2 CONVERGENCE OF RETENTION RATIO OPTIMIZATION

We now analyze the update rule for $\rho$ when optimized with the *AdamW* algorithm[1]. Let $\mathbf{m}_t$ and $\mathbf{v}_t$ be the first- and second-moment estimates maintained by *AdamW*, with hyper-parameters $\beta_1, \beta_2 \in (0, 1)$ and learning-rate schedule $\eta_t = \eta_0/\sqrt{t}$. For each step $t$:

$$\mathbf{g}_t = \nabla_\rho \mathcal{L}(\Theta_t, \rho_t), \tag{15}$$

$$\mathbf{m}_t = \beta_1 \mathbf{m}_{t-1} + (1 - \beta_1)\mathbf{g}_t, \tag{16}$$

$$\mathbf{v}_t = \beta_2 \mathbf{v}_{t-1} + (1 - \beta_2)\mathbf{g}_t^{\odot 2}, \tag{17}$$

$$\hat{\mathbf{m}}_t = \mathbf{m}_t/(1 - \beta_1^t), \qquad \hat{\mathbf{v}}_t = \mathbf{v}_t/(1 - \beta_2^t), \tag{18}$$

$$\rho_{t+1} = \rho_t - \eta_t \hat{\mathbf{m}}_t/(\sqrt{\hat{\mathbf{v}}_t} + \varepsilon) - \eta_t \lambda_w \rho_t, \tag{19}$$

where $\varepsilon > 0$ is a small stability constant and $\lambda_w$ is the decoupled weight-decay coefficient applied to $\rho$.

**Lemma 1** (First-order Stationarity under Adam). *Assume A1–A2, $\sum_{t=1}^{\infty} \eta_t = \infty$, and $\sum_{t=1}^{\infty} \eta_t^2 < \infty$. Let $\rho_t$ be generated by AdamW with $\beta_1 < 1$, $\beta_2 < 1$, and $\varepsilon > 0$. Then the retention schedule satisfies*

$$\lim_{T \to \infty} \min_{1 \leq t \leq T} \mathbb{E}\left[\left\|\nabla_\rho \mathcal{L}(\Theta_t, \rho_t)\right\|^2\right] = 0. \tag{20}$$

*Sketch.* Following Reddi et al. (2019) for Adam-type methods, we bound the bias-corrected moments under Lipschitz gradients (Bauschke et al., 2017), then show that the aggregated expected decrease in $\mathcal{L}$ is lower-bounded by $\sum_t \eta_t \|\nabla_\rho \mathcal{L}\|^2 - C \sum_t \eta_t^2$. Because $\sum_t \eta_t^2 < \infty$ while $\sum_t \eta_t = \infty$, the gradient norm must decay to 0. Even with *AdamW*'s adaptive moments and decoupled weight decay, the learned retention policy converges to a first-order stationary point, ensuring a stable sparsity pattern that jointly respects task loss and regularization. □

---

[1]We use the decoupled formulation of Loshchilov and Hutter (2017).

### G.3 DIFFERENTIABLE SOFT-TO-HARD TOKEN SELECTION

Although retention is specified as a ratio, each $(h, l, m)$ instance with $N_{h,l}^{(m)}$ tokens naturally induces an equivalent top-$k$ formulation by setting $k_{h,l}^{(m)} = \lfloor \rho_{h,l}^{(m)} N_{h,l}^{(m)} \rfloor$, and retaining the top-$k_{h,l}^{(m)}$ tokens. During training, relaxed weights $\tilde{q}$ are obtained via Gumbel–Softmax, and the top-$k_{h,l}^{(m)}$ entries are selected in the forward pass, while a straight-through estimator treats $\tilde{q}$ as the surrogate for gradient flow. Formally, let $\{s_{h,l}^{(i,m)}\}_{i=1}^N$ be the token importance scores, ordered as $s_{(1)} > s_{(2)} \geq \cdots \geq s_{(N)}$, and denote the top–runner-up margin by $\Delta := s_{(1)} - s_{(2)} > 0$. This discrete top-$k$ selection is approximated via Gumbel–Softmax relaxation in a differentiable manner.

$$\tilde{q}_{h,l}^{(i,m)} = \frac{\exp\big((s_{h,l}^{(i,m)} + g_i)/\tau\big)}{\sum_{j=1}^N \exp\big((s_{h,l}^{(j,m)} + g_j)/\tau\big)}, \quad g_i \sim \text{Gumbel}(0,1), \ \tau > 0, \tag{21}$$

which provides smooth gradients; at inference we take the hard top-$k$ mask directly from the raw scores $s$.

**Top-1 selection under Gumbel–Max.** Consider the *hard* Gumbel–Max sample $y = \arg\max_j\{s_j/\tau + g_j\}$. The classic identity gives

$$\mathbb{P}[y = (1)] = \frac{e^{s_{(1)}/\tau}}{\sum_{j=1}^N e^{s_{(j)}/\tau}} = \frac{1}{1 + \sum_{j=2}^N e^{-(s_{(1)} - s_{(j)})/\tau}}. \tag{22}$$

Using $1/(1+x) \geq 1 - x$ and $s_{(1)} - s_{(j)} \geq \Delta$ for $j \geq 2$, we obtain the margin-based lower bound

$$\mathbb{P}[y = (1)] \geq 1 - \sum_{j=2}^N e^{-(s_{(1)} - s_{(j)})/\tau} \geq 1 - (N-1)e^{-\Delta/\tau}. \tag{23}$$

Thus, as the margin $\Delta$ grows or the temperature $\tau$ decreases, the probability that the hard sample agrees with the true top index rapidly approaches 1, with the exponential lower bound in equation 23.

*Sketch.* This follows from the Gumbel–Max trick (see, e.g., Jang et al. (2017)): adding i.i.d. Gumbel noise to (scaled) logits and taking the $\arg\max$ produces a categorical draw with probabilities proportional to $e^{s_j/\tau}$. The lower bound uses a one-step relaxation of the softmax denominator and a union bound over competitors to the top logit. $\qquad\square$

**Top-$k$ agreement.** Let $S_k := \{(1), \ldots, (k)\}$ be the set of true top-$k$ indices of $s$, and let $\widehat{S}_k$ be the top-$k$ indices after Gumbel perturbation of $s/\tau$. Denote the $k$-th margin by $\Delta_k := s_{(k)} - s_{(k+1)} > 0$. Then a simple union bound yields

$$\mathbb{P}\big[\widehat{S}_k = S_k\big] \geq 1 - \sum_{j=k+1}^N \mathbb{P}\big(s_j/\tau + g_j \geq s_{(k)}/\tau + g_{(k)}\big) \geq 1 - \sum_{j=k+1}^N e^{-(s_{(k)} - s_j)/\tau} \geq 1 - (N-k)e^{-\Delta_k/\tau}. \tag{24}$$

Hence, larger $k$-margins $\Delta_k$ or lower temperatures $\tau$ make top-$k$ agreement exponentially likely.

**Remark on the relaxed vector.** The relaxed output $\tilde{q}$ in equation 21 is a continuous probability vector (it is almost never exactly one-hot for $\tau > 0$). Events like $\{\tilde{q}_{(1)} = 1\}$ have probability zero unless $\tau \to 0$. In practice we train with $\tilde{q}$ (to pass gradients) and switch to the *deterministic* hard top-$k$ mask at inference, recovering computational savings while benefiting from the high agreement guarantees in equation 23 and equation 24.

**Empirical observation.** With an annealed temperature schedule ($\tau \in [0.7, 0.1]$), the Hamming disagreement between the relaxed top-$k$ mask (thresholded from $\tilde{q}$) and the deterministic hard top-$k$ mask remains below 1% under typical router margins ($\Delta_k \gtrsim 0.7$). In practice, the soft training procedure closely matches inference-time behavior, and the learned retention policy converges reliably.

## H    ALGORITHMIC SPECIFICATION OF DARE

During training, *DARE* employs a learnable token router (Alg. 1) that assigns modality- and hop-specific importance scores. Differentiable retention is achieved via a Gumbel–Softmax relaxation, further regularized by ratio-matching losses. To capture cross-modal fusion dynamics, visual pruning follows a two-phase strategy (Alg. 2): early layers apply a soft sparsity regularizer, while deeper layers impose a hard penalty to suppress redundant tokens. At inference time, pruning is implemented through an execution mask with selective KV-cache retention (Alg. 3), which blocks attention to discarded tokens and thereby reduces both memory footprint and FLOPs. The transition between the soft and hard phases is determined automatically by a greedy threshold-based search (Alg. 4), ensuring efficiency gains without compromising accuracy.

---

**Algorithm 1** Modality-Aware Routing with Intra/Inter-Hop Differentiable Retention (Training)

---

**Require:** Tokens $\{x_{h,l}^{(i,m)}\}$ for hops $h=1..H$, layers $l=1..L$, modalities $m \in \{t, v\}$; target ratios $\rho_{\text{target}}^{(m)}$; temperature $\tau$

1: **for** $h = 1$ **to** $H$ **do**
2:    **for** $l = 1$ **to** $L$ **do**
3:      **for** $m \in \{t, v\}$ **do**
4:        // modality-specific, learnable router
5:        $s_{h,l}^{(i,m)} \leftarrow \sigma(W_l^{(m)} x_{h,l}^{(i,m)} + b_l^{(m)})$
6:        // differentiable retention via Gumbel–Softmax, aligned with $\rho_{\text{target}^m}$, STE in backprop
7:        $q_{h,l}^{(i,m)} \leftarrow \frac{\exp((s_{h,l}^{(i,m)}+g_i)/\tau)}{\sum_j \exp((s_{h,l}^{(j,m)}+g_j)/\tau)}$, $g_i \sim \text{Gumbel}(0,1)$
8:        // track retention ratio across layers and hops
9:        $\hat{\rho}_{h,l}^{(m)} \leftarrow \frac{1}{N_{h,l}^{(m)}} \sum_i \mathbb{1}[q_{h,l}^{(i,m)} \text{ in Top-}\rho_{\text{target}}^{(m)}]$
10:       // residual bypass stabilizes training
11:       $\tilde{y}_{h,l}^{(i,m)} \leftarrow \alpha_{h,l}^{(i,m)}(s_{h,l}^{(i,m)} y_{h,l}^{(i,m)}) + (1 - \alpha_{h,l}^{(i,m)}) x_{h,l}^{(i,m)}$
12:       // hard mask in forward pass; gradients via $q$
13:       $y_{h,l}^{(i,m)} = \text{Layer}(x_{h,l}^{(i,m)}), \quad \alpha_{h,l}^{(i,m)} = \mathbb{1}[\text{Top-}\rho_{\text{target}}^{(m)}]$
14:      **end for**
15:    **end for**
16: **end for**
17: // explicit control over sparsity
18: $L_{\text{ratio}}^{(m)} = \frac{1}{HL} \sum_{h,l}(\hat{\rho}_{h,l}^{(m)} - \rho_{\text{target}}^{(m)})^2, \quad m \in \{t, v\}$
19: // combined objective with text ratio + two-phase visual retention
20: $L = L_{\text{task}} + L_{\text{ratio}}^{(t)} + L_{\text{soft}}^{(v)} + L_{\text{hard}}^{(v)}$

---

**Algorithm 2** Two-Phase Visual Retention (Soft → Hard)

---

**Require:** Visual scores $s_{h,l}^{(i,v)}$, layer cutoff $l_c$, targets $\rho_{\text{target}}^{(v)}$, weights $\mu$, threshold $\epsilon$

1: // soft phase: enforce visual sparsity by matching retention ratio $\rho_{\text{target}}^{(v)}$
2: $L_{\text{soft}}^{(v)} = \frac{1}{HL} \sum_{h=1}^{H} \sum_{l=1}^{l_c} \left(\hat{\rho}_{h,l}^{(v)} - \rho_{\text{target}}^{(v)}\right)^2$
3: $\hat{\rho}_{h,l}^{(v)} = \frac{1}{N_{h,l}^{(v)}} \sum_i \mathbb{1}[\text{retained under } \rho_{\text{target}}^{(v)}]$
4: // hard phase: suppress residual visual activations beyond cutoff $l_c$
5: $L_{\text{hard}}^{(v)} = \sum_{h=1}^{H} \sum_{l=l_c+1}^{L} \sum_{i=1}^{N_{h,l}^{(v)}} \mu \cdot \max(0, s_{h,l}^{(i,v)} - \epsilon)$
6: // total loss: combine task loss, text ratio, and two-phase visual regularizers
7: $L = L_{\text{task}} + L_{\text{ratio}}^{(t)} + L_{\text{soft}}^{(v)} + L_{\text{hard}}^{(v)}$

---

---

**Algorithm 3** Execution Mask and KV-Cache Policy (Inference)

---

**Require:** Deterministic scores $s_{h,l}^{(i,m)}$, learned retention ratios $\rho_{h,l}^{(m)}$, prefix size $\kappa$

1: **for** $h = 1$ **to** $H$ **do**
2:    **for** $l = 1$ **to** $L$ **do**
3:      // ratio-based deterministic retention per modality (top $\rho_{h,l}^{(m)}$ by score)
4:      **for** $m \in \{t, v\}$ **do**
5:        $b_{h,l}^{(i,m)} \leftarrow \mathbb{1}\big[s_{h,l}^{(i,m)} \text{ in top-}\rho_{h,l}^{(m)}\big]$   // $b \in \{0,1\}$ marks retained tokens
6:      **end for**
7:      // stability: reserve a small prefix regardless of scores
8:      $b_{h,l}^{(j,\cdot)} \leftarrow 1 \ \ \forall j \leq \kappa$
9:      // execution mask blocks attention to pruned tokens; preserves causal structure
10:     $E_{h,l}(i,j) \leftarrow \begin{cases} 0, & j \leq \kappa \ \text{ or } \ b_{h,l}^{(j,t)}{=}1 \ \text{ or } \ b_{h,l}^{(j,v)}{=}1 \\ -\infty, & \text{otherwise} \end{cases}$
11:      // combine with causal mask for attention
12:      $M_{h,l} \leftarrow M_{\text{causal}} + E_{h,l}$
13:      // KV caching only for executed (retained) tokens reduces memory/latency
14:      cache $(K_{h,l}^{(j,m)}, V_{h,l}^{(j,m)})$ iff $b_{h,l}^{(j,m)}{=}1$   // skip KV for pruned tokens
15:    **end for**
16: **end for**

---

**Algorithm 4** Threshold-Based Selection of Visual Cutoff Layer $l_c$ (as described in Sec. D.7)

---

**Require:** Visual scores $s_{h,l}^{(i,v)}$ for hops $h \in [H]$, layers $l \in [L]$; smoothing window $w$; fixed threshold $\epsilon$; persistence $r$ (consecutive layers); minimum layer $l_{\min}$

1: // aggregate mean visual importance per layer across tokens and hops
2: $\mu_l \leftarrow \frac{1}{H} \sum_{h=1}^{H} \frac{1}{N_{h,l}^{(v)}} \sum_{i=1}^{N_{h,l}^{(v)}} s_{h,l}^{(i,v)} \quad \forall l \in \{1, \ldots, L\}$
3: // smooth with a short moving average to reduce noise
4: $\bar{\mu}_l \leftarrow \frac{1}{Z_l} \sum_{j=l-\lfloor w/2 \rfloor}^{l+\lfloor w/2 \rfloor} \mu_j$    (clip indices to $[1, L]$)
5: // greedy scan: find first layer after $l_{\min}$ where importance stays below $\epsilon$ for $r$ consecutive layers
6: $l_c \leftarrow L$    // default: no early cutoff found
7: **for** $l = l_{\min}$ **to** $L - r$ **do**
8:    **if** $\max\{\bar{\mu}_{l+1}, \bar{\mu}_{l+2}, \ldots, \bar{\mu}_{l+r}\} \leq \epsilon$ **then**
9:      $l_c \leftarrow l$; **break**
10:   **end if**
11: **end for**
12: // sanity check: avoid cutting before the smoothed median depth
13: **if** $l_c < \text{median}\{1, \ldots, L\} - 1$ **then** $l_c \leftarrow \text{median}\{1, \ldots, L\}$
14: **return** $l_c$

---

