# OpenReview forum: "Efficient Multimodal Spatial Reasoning via Dynamic and Asymmetric Routing"
_ICLR.cc/2026/Conference — ICLR 2026 Poster_

### Official Review · Reviewer_ddYw · 2025-10-26

**Soundness:** 2
**Presentation:** 2
**Contribution:** 1
**Rating:** 4
**Confidence:** 3

**Summary:**

This paper introduces DARE (Dynamic and Asymmetric Routing), a new framework designed to address the significant computational and memory costs associated with multi-hop reasoning in Multimodal Large Language Models (MLLMs). It proposes a dynamic token pruning mechanism. Through experiments on seven spatial reasoning benchmarks, the authors demonstrate that DARE achieves substantial reductions in FLOPs and KV-cache usage.

**Strengths:**

The paper provides a clear motivation by outlining the computational efficiency challenges in multi-hop multimodal reasoning. Furthermore, the work is presented with a coherent structure, making the methodology and experimental sections straightforward to follow.

**Weaknesses:**

Regarding the experiments on dynamic spatial reasoning, I am genuinely interested in understanding how the results in Table 2 were obtained. As someone familiar with unified MLLMs and interleaved multimodal reasoning, I noticed that the datasets mentioned—MAZE, MINIBEHAVIOR, and FROZENLAKE—do not appear to be publicly available, at least to the best of my knowledge. This raises some concerns about the reproducibility and authenticity of the reported results, as independent verification is currently not feasible. I believe that providing access to the source code and implementation details of this part would help clarify how the experiments were conducted. To adhere to the double-blind review policy, an anonymous link to a code repository would be very helpful.

**Questions:**

Please see the weaknesses above.

---

> ### Author Response · Authors · 2025-11-21
> **Reproducibility of Dynamic Spatial Reasoning Results and Dataset Reconstruction (Q1 Response)**
>
> We sincerely thank the reviewer for the thoughtful evaluation. We are grateful for the positive recognition that DARE provides a well-motivated and new solution to the computational and memory challenges inherent in multi-hop multimodal reasoning. We also appreciate the reviewer’s acknowledgement that the paper presents the new framework, solid methodology, and experimental results in a coherent and accessible manner.
>
>
> **`Q1: Dynamic Spatial Reasoning Results`**
>
>  **`A: `** We appreciate the reviewer’s close engagement with multimodal reasoning systems and the thoughtful question regarding how the dynamic spatial reasoning results in Table 2 were obtained, given that the original MVoT's datasets and checkpoints are not publicly released. Below we clarify the full process and provide concrete reproducibility guarantees.
>
> The key point is that MAZE, MiniBehavior, and FrozenLake are not static datasets, but *procedurally generated benchmarks* built on top of *publicly available environment codebases*. The MVoT paper[1] clearly describes their sampling rules, intermediate-thought formats, and environment dynamics in detail, which makes all three tasks reproducible from first principles. In our work, we do not use any private data; rather, all trajectory datasets are re-generated deterministically using publicly available codebases and the specifications provided in MVoT.
>
> To ensure complete transparency, we provide an anonymous repository(**https://anonymous.4open.science/r/DARE-anonymous-0C77**), which is also included in the `supplementary material`, with end-to-end scripts for:
>
> (1) **generating the MAZE, MiniBehavior, and FrozenLake datasets using the publicly available environment codebases** (each generator is deterministic and outputs the exact interleaved multimodal trajectories used in our experiments)
>
> (2) **recovering the baseline MVoT-style interleaved reasoning by fine-tuning Anole-7B on these datasets**, following the hyperparameters and reasoning format specified in the MVoT paper; for transparency, we also include a small portion of the generated data at: `/data-samples.zip` in **https://anonymous.4open.science/r/DARE-anonymous-0C77**.
>
> (3) **training and evaluating DARE using DeepSpeed ZeRO-3 on 4×A100-40GB nodes**. The repository includes the dataset generators, environment wrappers, prompt templates, training configuration files, DeepSpeed launcher scripts, and the DARE implementation with importance scoring, differentiable top-k routing (Gumbel-soft relaxation), and gather-based KV-cache pruning for inference-time speedups.
>
> * For MAZE dataset [2], we use the official maze-dataset library
>   (*https://github.com/understanding-search/maze-dataset*),
> we implement a generator that samples maze sizes 3–6 with different seeds, computes valid paths, renders each intermediate step with red-arrow visual thoughts (using updated utilities in `plot_maze.py`, detailed in **https://anonymous.4open.science/r/DARE-anonymous-0C77/maze_datagenerator/README.md**), and removes repeated trajectories to prevent train–dev leakage. A dataset slice is included for reference.
>
> * For the MiniBehavior dataset [3], we build on the official mini_behavior codebase
>  (*https://github.com/StanfordVL/mini_behavior*).
>   We implement a complete dataset generator (`generate_ppo_multilevel_dataset.py` in `anonymous-0C77/minibehavior_datagenerator`). A PPO agent (Stable-Baselines3) is trained in the *INSTALLINGAPRINTER environment* for room sizes 7, 8, 9, and 10; for reproducibility, we provide the exact room-size-7 model used in our experiments (see **https://anonymous.4open.science/r/DARE-anonymous-0C77/minibehavior_datagenerator/README.md**). The generator then extracts successful action sequences along with all rendered intermediate frames. Both the generator code and a representative data subset are included in the anonymous repository.
>
> * For the FrozenLake dataset [4], we use the Visual Spatial Planning environment (*https://github.com/UCSB-NLP-Chang/Visual-Spatial-Planning*), we implement `frozen_lake_unified_balance.py` to collect Q-table–guided trajectories, following the success/failure balancing rule described in MVoT. All intermediate visual thoughts are rendered in the same format as MVoT, as detailed in **https://anonymous.4open.science/r/DARE-anonymous-0C77/frozenlake_datagenerator/README.md**.  Both the generator script and a representative dataset sample are included in the anonymous repository.
>
> [1] Li, Chengzu, et al. *Imagine while reasoning in space: Multimodal visualization-of-thought* (ICML 2025).
>
> [2] Ivanitskiy et al. *A configurable library for generating and manipulating maze datasets* (in Hugging Face
>  2023).
>
> [3] Jin et al. *Mini-Behavior: A procedurally generated benchmark for long-horizon decision-making in embodied AI* (NeurIPS workshop 2023).
>
> [4] Wu et al. *VSP: Assessing the dual challenges of perception and reasoning in spatial planning tasks for MLLMs* (ICCV 2024).

---

### Official Review · Reviewer_wQMw · 2025-10-28

**Soundness:** 3
**Presentation:** 4
**Contribution:** 4
**Rating:** 8
**Confidence:** 4

**Summary:**

This paper proposes a new framework that enhances the efficiency of MLLMs during spatial-reasoning tasks by adaptively pruning multimodal tokens across network depths, reasoning hops, and modalities. DARE integrates a layer- and hop-aware differentiable retention mechanism with an asymmetric compression strategy to selectively preserve essential tokens. Experiments on multiple multimodal spatial reasoning benchmarks show that DARE significantly reduces FLOPs and memory overhead while maintaining performance.

I have reviewed this paper for NeurIPS 2025. After careful check, I noticed that the authors have incorporated the reviewers' suggestions and addressed their concerns in this version. For example, the comparison between DARE and MoD (Appendix D.4: Comparison with Heuristic Token Retention and Routing Methods) and the experiments across models of different sizes (Scalability to Larger and Smaller Models, Section G.1). I think this version is clear and well-written, meeting the standard for acceptance.

**Strengths:**

1. The motivation is solid, and the method is well-motivated. The authors tackle a key efficiency bottleneck in multi-hop multimodal spatial reasoning. The approach is clearly formulated and effectively presented.

2. The main experiments are comprehensive, covering seven spatial-reasoning benchmarks, general VQA, hallucination detection, and detailed ablation studies.

3. The authors provide thorough discussion and analysis, including comparisons between DARE and MoD, interpretation of retention rates, the necessity of learning-based routing, and how DARE scales as model size increases. Overall, the paper offers a complete and clear discussion of the DARE method and its position within the broader research landscape.

**Weaknesses:**

No obvious weaknesses.

**Questions:**

After careful review, the authors have addressed all the questions I raised in my previous review, and I have no further questions.

---

> ### Author Response · Authors · 2025-11-21
> **Appreciation for Your Positive Recognition and Remarks**
>
> We sincerely thank the reviewer for the clear and supportive evaluation of our revised submission. We greatly appreciate the recognition that the additional analyses, such as the comparison with MoD in Appendix D.4 and the scalability results in Section G.1, effectively addressed the concerns raised in the earlier round.
>
> Your positive assessment of the paper’s motivation, methodological clarity, and comprehensive experimental coverage is deeply encouraging. We are grateful for the reviewer’s careful re-examination and for acknowledging that the revised version meets the standard for acceptance. Thank you for the thoughtful and constructive feedback throughout the review process.

---

### Official Review · Reviewer_8DKA · 2025-10-30

**Soundness:** 4
**Presentation:** 3
**Contribution:** 3
**Rating:** 6
**Confidence:** 3

**Summary:**

This paper introduces DARE, a novel framework designed to improve the efficiency of multi-hop multimodal spatial reasoning by addressing the high computational costs of existing methods. It achieves this through several key contributions: a modality-aware token routing mechanism with dedicated routers at each transformer layer to score token importance; an intra- and inter-hop aware retention strategy that uses Gumbel-Softmax to learn dynamic pruning ratios across both network layers and reasoning steps; and a KV-cache policy that reduces memory overhead by aligning with token pruning decisions. The authors report significant efficiency gains while maintaining or improving performance.

**Strengths:**

1. DARE shows compelling performance, consistently achieving comparable or superior accuracy to its baselines while being significantly more efficient.
2. The framework demonstrates impressive generalization by showing effectiveness on two different interleaved reasoning architectures: one based on visual token referencing (VolCano) and another on mental image generation (Anole).
3. The paper is supported by a comprehensive set of experiments and ablation studies that strongly validate the proposed design choices. The authors' effort in providing this level of detail, including a well-organized appendix, is very impressive.

**Weaknesses:**

1. **Impact of New Hyperparameters on Applicability:** The proposed method introduces several new hyperparameters (e.g., target retention ratios $p_{target}$, hard pruning threshold $\epsilon$, prefix size $\kappa$). While the paper includes helpful ablation studies, finding the optimal set of these hyperparameters for a new model or task could be a non-trivial and expensive process, potentially limiting the method's plug-and-play applicability.

2. **Fixed Number of Reasoning Hops**: The framework appears to operate with a pre-defined maximum number of reasoning hops ($H$). Real-world problems may require a variable number of reasoning steps. It is unclear how DARE would adapt to tasks that require more or fewer hops than what it was trained for, which may limit its flexibility in more open-ended scenarios.

**Questions:**

1. How do the router's importance scores change when the model is given a different task? For instance, does a visual search task yield a different tendency compared to a dynamic navigation task (like MAZE)?

2. The term "hop" is central to the paper but is not explicitly defined. Is a "hop" defined as a single autoregressive generation step for an intermediate thought (which could be a mix of visual and textual tokens), as suggested by Figure 2? A precise definition would be helpful.

3. In Table 1, DARE-LH shows a particularly large accuracy improvement over the VolCano baseline on EmbSpatial compared to other compositional tasks like VSR. Is there an intuition for why the proposed method is especially effective for this specific benchmark?

4. Table 3 reports results for DARE-L but not DARE-LH on the General VQA and Hallucination benchmarks. Is this because these tasks are typically solved in a single reasoning step (i.e., one hop), making the inter-hop (-LH) mechanism not applicable?

---

> ### Author Response · Authors · 2025-11-21
> **Practical Applicability of DARE’s Hyperparameters (W1 Response)**
>
> We sincerely thank the reviewer for the thoughtful and constructive assessment of our work. We greatly appreciate the recognition of DARE’s strong empirical performance, its effective generalization across distinct interleaved reasoning architectures, and the clarity and depth of our experimental analysis. The reviewer’s positive remarks regarding the soundness of the methodology and the quality of the appendix are deeply encouraging. We are also grateful for the insightful questions and concerns, which have helped us substantially clarify the framework’s applicability, the definition of key concepts, and the intuitions behind several design choices. We have incorporated this analysis into the revised paper (Lines 513–529) and provide detailed responses below.
>
> **`W1: Impact of New Hyperparameters on Applicability`**
>
>
>  **`A: `** We thank the reviewer for raising this important point. Although DARE introduces several hyperparameters (ρᵛ, ρᵗ, τ, κ, ε), the newly added radar sensitivity analysis in Fig. 7 shows that their practical tuning burden is minimal. Our ablations further confirm this through three observations detailed below.
>
> (1) **Most hyperparameters exhibit wide stability regions.**
> Across all benchmarks, the global retention ratios (ρᵛ and ρᵗ) are the only hyperparameters that meaningfully influence the accuracy–efficiency trade-off. All other parameters (κ, ε, l_c, τ) cause less than a one-point variation in accuracy, demonstrating that DARE does not require fine-grained adjustment. In practice, users primarily tune a *single knob*, the global retention ratio, to balance sparsity and performance.
>
> (2) **A single default configuration generalizes across models and tasks.**
> The same settings (ρᵛ=0.4, ρᵗ=0.7, τ=0.7, κ=2, ε=0.01) work robustly for both VolCano-7B and Anole-7B and across all seven benchmarks. The automatically identified cutoff depth l_c further removes the need for manual tuning. Furthermore, the radar sensitivity analysis introduced in the revision (Lines 513–519) shows that varying only the global retention ratios provides a smooth and predictable trade-off between efficiency and accuracy.
>
> (3) **End-to-end learning eliminates the need for complex schedules.**
> Unlike methods (e.g., Heima) that require multi-stage curricula or extensive hyperparameter optimization, DARE trains in a single 10-epoch adaptor stage. Sparsity patterns are learned end-to-end under global sparsity targets, so users typically only specify the desired retention level, and the routing mechanism automatically adjusts the remaining pruning behaviors.
>
>
> Table A4 provides a consolidated summary of all hyperparameters used in DARE, along with their roles and empirical sensitivity. We have incorporated this analysis into the revised paper (Lines 513–529).
>
> Table A4 Summary of DARE’s hyperparameters, their roles, and empirical sensitivity.
>
> | Category             | Hyperparameter            | Value Used    | Role in DARE                                    | Sensitivity Observation                        |
> | -------------------- | ------------------------ | ------------- | ----------------------------------------------- | ---------------------------------------------- |
> | Retention targets    | Visual retention ratio ρᵛ | 0.4          | Controls fraction of visual tokens retained     | Main sensitivity knob; deviation ≤1.06%        |
> |                      | Text retention ratio ρᵗ   | 0.7          | Controls fraction of textual tokens retained    | Main sensitivity knob; deviation ≤1.00%        |
> | Routing temperature  | Gumbel–Softmax τ          | 0.7           | Smoothness of differentiable routing scores     | Stable across τ ∈ [0.3, 1.3]; ≤0.8–1.0% impact |
> | KV-cache routing     | Prefix length κ           | 2             | Small prefix preserved for decoding stability   | Stable for κ ≥ 2; ≤0.05% deviation             |
> | Visual pruning       | Visual cutoff depth l_c   | auto-detected | Determines transition from soft to hard pruning | ±3 layer shift changes ≤0.1%                   |
> | Threshold parameters | Pruning threshold ε       | 0.01          | Suppresses late-layer visual tokens             | Very robust; ≤0.2–0.3% deviation               |

---

> ### Author Response · Authors · 2025-11-21
> **Handling Variable-Length Reasoning and Hop Flexibility (W2 Response)**
>
> **`W2: Fixed Number of Reasoning Hops`**
>
>  **`A: `** Thank you for highlighting this question. We apologize for any ambiguity in our initial wording. We clarify that *DARE is not tied to any fixed maximum number of reasoning hops*; the *H = 5 hops* shown in Fig. 1 and Fig. 3 serves only as an illustrative example and does not reflect a limitation of the framework. In practice, DARE is trained and evaluated on tasks that naturally involve *variable-length multimodal reasoning*. The average numbers of intermediate steps are 9.11 for MAZE, 7.83 for MiniBehavior, and 6.56 for FrozenLake, with substantial episode-to-episode variation. To make this explicit, we added Fig. 6 in the revision (Lines 491–511), which shows that different MAZE layouts naturally give rise to trajectories of different lengths (e.g., 3, 5, or 7 hops).
>
> This flexibility is enabled by three design choices that allow DARE to naturally accommodate variable-length reasoning sequences:
>
> (1) **DARE’s routing decisions are entirely local to each hop**, which means that each hop is processed without referencing any global rollout information. At hop (h) and layer (l), the router takes only the current multimodal hidden state ($Z_{h,l}$) as input and produces a hop-specific importance distribution and token-retention mask. Crucially, this computation does not depend on the total number of hops executed so far, the remaining number of hops, or any pre-specified maximum horizon. Because routing is conditioned solely on ($Z_{h,l}$), multimodal thoughts that involve fewer hops than those seen during training are handled directly: the model simply halts when the reasoning sequence ends, and DARE applies its hop-local rule to the hops that actually occur. Multimodal thoughts that require more hops than training are equally supported: the routing parameters are shared across hops, and the retention behavior for additional hops is governed by the corresponding target ratios ($\rho_{h,l}$).
>
>
>
> (2) **Pruning strength is governed by adaptive token-importance signals rather than a fixed hop schedule.**
> The model detects semantic transitions through changes in router scores and adjusts retention at each hop accordingly. This enables pruning decisions to follow the content of the intermediate thought, not its position in a predetermined sequence. In practice, this mechanism absorbs the natural variability in reasoning patterns that arises in MAZE, MiniBehavior, and FrozenLake, where the number and nature of intermediate thoughts differ across episodes.
>
>
>
> (3) **DARE is trained on non-uniform intermediate-thought trajectories and therefore learns routing behaviors that generalize across reasoning depths.**
> Since the training data includes trajectories with diverse hop lengths, the model learns a hop-local routing rule that extrapolates naturally to sequences of different lengths. Longer sequences simply apply the learned rule over more iterations, while shorter sequences terminate without disrupting the architecture. For hops beyond the training horizon, we reuse the stable late-hop targets learned during training, which empirically capture the correct pruning pattern.

---

> ### Author Response · Authors · 2025-11-21
> **Task Adaptation of Routing Behavior and Definition of Reasoning Hops (Q1+Q2 Responses)**
>
> **`Q1: How the Router’s Importance Scores Adapt Across Different Tasks`**
>
>  **`A: `** Thank you for this insightful question. To directly evaluate whether DARE’s routing behavior adapts to different tasks, we conducted an additional cross-task analysis. In the main paper, VSR is evaluated on VolCano-7B (Table 1), while MAZE is evaluated on Anole-7B (Table 2). To isolate task effects from architectural effects, we further ran VSR on Anole-7B using the *same DARE adaptor trained exclusively on navigation tasks*. This setup allows us to observe how the router behaves when the task shifts from dynamic multi-hop navigation to perception-centric visual search.
>
> We find that routing patterns differ substantially *before* the information-cliff point.
> For MAZE, the router progressively shifts importance toward textual tokens across hops because correct planning requires accumulating and integrating intermediate thoughts. In contrast, VSR on Anole-7B exhibits a different trend: the router retains significantly more early-layer visual tokens and prunes text more aggressively, consistent with VSR’s emphasis on fine-grained perception and spatial grounding. This demonstrates that the router does not reuse the MAZE pattern; rather, it adjusts its importance scores according to the task demands, even under the same global sparsity targets (ρᵛ = 0.4, ρᵗ = 0.7).
>
> Importantly, *after* the curvature break (the information-cliff described in Appendix D.7), both tasks converge to similar behavior. Once the router detects the information-fusion point, the curvature-based heuristic triggers the soft-to-hard transition, and late-layer visual tokens are pruned deterministically. This convergence occurs despite differences in tasks, datasets, and backbones, indicating that the hard-pruning phase is governed by the input’s information-flow structure rather than by memorized task-specific patterns.
>
>
> Quantitatively, applying the MAZE-trained Anole adaptor to VSR produces only a small accuracy drop and a modest increase in FLOPs compared to running on VolCano-7B (Table A5). Accuracy decreases slightly (68.09% → 67.41%), FLOPs increase moderately (11,310G → 12,893G), and latency remains nearly unchanged (0.41s → 0.42s). Despite the backbone change and the task mismatch, the adaptor still identifies the correct information-cliff and applies hard pruning at the appropriate depth.
>
>
> Table A5 VSR on VolCano-7B vs. VSR on Anole-7B
>
> | Model + Setting                     | Accuracy | FLOPs (G) |  Latency (s) |
> | -------- | --------- | ---| ----------- |
> | VolCano-7B + DARE-LH| 68.09%   | 11,310                | 0.41        |
> | Anole-7B + DARE-LH | 67.41%   | 12,893               | 0.42        |
>
>
>
>
> **`Q2: Definition of “Hop”`**
>
>  **`A: `** Thank you for this very helpful suggestion. We agree that the notion of a “hop” should be stated explicitly, and we have clarified this in the revised version (Lines 157–161).
>
> In our framework, a *hop* corresponds to one complete autoregressive reasoning cycle in which the model generates both an intermediate visual thought and an intermediate textual thought, as depicted in Figure 2. Following Eq. (1), the model first produces a visual thought conditioned on the input and all previously generated thoughts, and then generates the corresponding textual thought conditioned on the updated history. This visual–textual pair constitutes a single hop in the overall multimodal reasoning trajectory.

---

> ### Author Response · Authors · 2025-11-21
> **Performance Differences on EmbSpatial and Applicability of DARE-LH to Single-Hop Tasks (Q3+Q4 Responses)**
>
> **`Q3: Why DARE-LH Shows Larger Gains on EmbSpatial?`**
>
>
>  **`A: `** Thank you for highlighting this observation. We investigated this behavior and found that the larger improvement on EmbSpatial arises naturally from how its task characteristics interact with DARE’s intra-/inter-hop routing and asymmetric compression. Three factors contribute to this effect.
>
> (1) **EmbSpatial requires deeper cross-modal fusion, making it more sensitive to preserving early visual evidence.**
>
> EmbSpatial queries depend on multiple object-level spatial cues, so fine-grained visual information must be preserved across more layers before being fully integrated into the textual stream. As illustrated in Fig. 1, visual-token importance remains substantial in the early and mid layers and only decreases once the relational structure has been encoded linguistically. *DARE’s layer-adaptive routing follows this pattern: it retains visual tokens during the layers where spatial grounding is still being built, and prunes only after these cues have been absorbed. This prevents the premature loss of entity-level evidence, a failure mode more likely in VolCano and less critical for simpler tasks like VSR, where grounding relies on a single relation and far fewer visual details.*
>
> (2) **EmbSpatial produces longer and denser multimodal sequences, which intensify the impact of redundant tokens.**
>
> Because EmbSpatial scenes contain more objects and richer spatial descriptions, each example generates significantly more visual and textual tokens than VSR. Across multiple reasoning hops, these tokens accumulate quickly, increasing attention noise, expanding the KV cache, and reducing effective reasoning depth. *DARE’s routing and progressive KV retention counteract this by removing low-utility visual tokens early and preventing redundant tokens from propagating across hops. Since EmbSpatial’s sequences are substantially denser, this controlled reduction yields a disproportionately larger accuracy gain compared to VSR.*
>
> (3) **EmbSpatial exhibits a larger redundancy gap between informative and non-informative tokens, which DARE exploits effectively.**
>
> EmbSpatial scenes contain many background regions and distractor objects, while only a small subset of tokens correspond to the entities and relations that determine the answer. This creates a pronounced redundancy gap: most visual tokens carry little signal for the final prediction, unlike in VSR where a larger fraction of tokens are directly relevant. Uniform or symmetric pruning schemes either risk dropping these few critical entity tokens or end up keeping a large amount of irrelevant content. In contrast, *DARE’s intra-/inter-hop router assigns higher retention to relationally important visual anchors and prunes low-utility background tokens more aggressively, producing a more focused and semantically coherent representation. This targeted pruning is particularly beneficial on EmbSpatial, where performance hinges on preserving a small set of precise relational cues.*
>
> **`Q4: Why Table 3 Reports DARE-L Instead of DARE-LH?`**
>
>  **`A: `** Thank you for the thoughtful question. We clarify that this choice reflects the nature of the benchmarks rather than any limitation of DARE-LH. The General VQA and Hallucination datasets are typically solved in a single reasoning step (i.e., one hop), where the model directly predicts the answer without generating intermediate thoughts; as a result, the hop-adaptive mechanism in DARE-LH naturally collapses to the same behavior as DARE-L. Introducing multi-hop prompting in this setting would artificially create additional hops that these tasks were never designed for, deviating from standard single-shot evaluation protocols used in prior work (e.g., GQA, MMBench, POPE, AMBER). Such modification would inflate sequence length, introduce extra generated tokens unrelated to the task, and make comparisons unfair. For these reasons, we follow the established evaluation setting and report DARE-L, while noting that DARE-LH remains fully applicable but simply offers no additional effect when the task itself is single-hop.

---

### Official Review · Reviewer_UNBZ · 2025-10-31

**Soundness:** 2
**Presentation:** 3
**Contribution:** 2
**Rating:** 4
**Confidence:** 2

**Summary:**

This paper introduces DARE, a novel and well-motivated framework for efficient multimodal spatial reasoning. By dynamically and asymmetrically pruning tokens across layers and reasoning hops, the method achieves reductions in computation and memory while often improving task performance. The paper is supported with comprehensive analysis and evaluation with comparisons over different system variants.

**Strengths:**

1. The paper is well-motivated. The paper excels at identifying and motivating two nuanced challenges in multi-hop multimodal reasoning: the dynamic, shifting importance of tokens across both network depth and reasoning steps, and the asymmetric redundancy patterns between visual and textual data.
2. Interesting technical design. I really like the use of a differentiable, end-to-end learnable routing mechanism compared to fixed heuristics. Furthermore, the asymmetric compression strategy and the explicit focus on reducing KV-cache usage are critical for autoregressive models.
3. Comprehensive empirical evaluation. The authors validate DARE across an extensive and diverse set of benchmarks, comparing it against diverse baselines. The results are good, showing that DARE not only meets its efficiency goals in terms of FLOPs and KV cache reduction but also even improves accuracy, demonstrating the method's effectiveness.

**Weaknesses:**

1. Reproducibility and reliability of the results: I'm concerned with the experimental results in Table 2, especially with MVoT and VoT. To be more specific, I would appreciate it if the authors can clarify:
* Computational resources: can Anole 7B be trained with 40 GB GPUs for how long? Could the authors provide training logs for clarification purposes?
* Experiment results: given that there is no public available datasets and model checkpoints MVoT uses in the original paper, and the reported numbers in Table 2 strictly aligns with the results in MVoT paper, plus the experimental settings (number of epochs, types of GPUs, numbers of GPUs) are not same as in the original paper, I would appreciate the authors provide more details regarding this. (I tried to look into the supplementary materials in the zip file, but didn't find the training script)
2. Methodological complexity and hyperparameters. While effective, DARE is a complex system with numerous interacting components and hyperparameters (target retention ratios, pruning thresholds, Gumbel temperature, etc.). The paper could benefit from a clearer discussion of the tuning effort required and the generalizability of the chosen hyperparameter values to different model architectures or domains.

**Questions:**

See as above in weaknesses. I'm happy to adjust the scores if the author can address the concerns in the weaknesses.

Typo: L037 MoT should be MVoT

---

> ### Author Response · Authors · 2025-11-21
> **Details on Training Resources, Runtime, and Reproducibility (Q1.1 Response)**
>
> We sincerely appreciate your thoughtful and encouraging evaluation of the paper, particularly your recognition of DARE’s novelty, well-motivated formulation, technically interesting design, and comprehensive empirical study. Below, we provide detailed responses addressing each of the concerns you raised.
>
>  **`Q1.1: Computational resources`**
>
>  **`A: `** Thank you for carefully examining the computational resources required for our experiments. We provide full clarification below:
>
> (1) All results in Table 2 were obtained using 4× A100-40GB GPUs, and Anole-7B can indeed be fine-tuned within this memory budget. We consistently applied DeepSpeed ZeRO-3, activation checkpointing, and bfloat16 precision for all computations across MAZE, MiniBehavior, and FrozenLake.
>
> (2) The exact wall-clock training times are reported in Table A1, MAZE: 30 h 00 m 27 s, MiniBehavior: 82 h 40 m 14 s, and FrozenLake: 116 h 33 m 11 s. The variation across tasks reflects differences in trajectory length, environment stochasticity, and episode complexity, which affect forward-pass cost and number of intermediate reasoning hops.
>
> (3) We have released the anonymized training logs and complete training scripts in both the `supplementary material` and the anonymous code repository:
> **https://anonymous.4open.science/r/DARE-anonymous-0C77**.
> System-specific identifiers (e.g., usernames, filesystem roots, and cluster names) are replaced with placeholders such as < USER >, < FS_ROOT >, and < CLUSTER_PROVIDER > to comply with the double-blind review policy.
>
> Table A1: Wall-Clock Training Time for Anole-7B on Dynamic Spatial Reasoning Tasks
>
> | Task          | Train Set Size | Dev Set Size | Epochs | Wall-Clock Time   |
> |---------------|---------------|--------------|--------|----------------|
> | MAZE          | 5,007          | 1,255        | 40     | 30 h 00 m 27 s
> | MiniBehavior  | 6,400          | 1,604        | 40     | 82 h 40 m 14 s
> | FrozenLake    | 6,846          | 1,664        | 40     | 116 h 33 m 11 s

---

> ### Author Response · Authors · 2025-11-21
> **Details on Experimental Reproduction and Dataset Reconstruction (Q1.2 Response)**
>
> **`Q1.2 Experimental Results and Dataset Availability`**
>
> **`A:  `** Thank you for the thoughtful questions. We clarify below how the MVoT[1] / VoT[2] baselines were reproduced and how the datasets were reconstructed.
>
> (1) Reproducing the MVoT/VoT reasoning setup
>
> Because the MVoT paper does not release model checkpoints or datasets, evaluating DARE requires faithfully reproducing the MVoT protocol. Anole-7B does not natively support interleaved multimodal reasoning, so we (i) fine-tune Anole-7B following the MVoT specification, including the visual–textual interleaving structure, trajectory format, intermediate-thought rendering, and evaluation pipeline, and (ii) ensure that the reproduced model matches the MVoT's reasoning performance in Table 2, which then serves as the reference baseline for evaluating DARE.
>
> (2) Reconstructing the three datasets
>
> Although the original datasets are not public, the MVoT paper clearly describes the sampling rules, environment configurations, and intermediate-thought formats in detail. Each task also has an open-source environment implementation, which we extend with our own generators. All generators and usage instructions are provided in the anonymous GitHub repository, along with a portion of each dataset (`/data-samples.zip`) for verification in **https://anonymous.4open.science/r/DARE-anonymous-0C77**.
>
>
> * Maze dataset [3]. Using the official maze-dataset library
>   (*https://github.com/understanding-search/maze-dataset*),
> we implement a generator that samples maze sizes 3–6 with different seeds, computes valid paths, renders each intermediate step with red-arrow visual thoughts (using updated utilities in `plot_maze.py`, detailed in **https://anonymous.4open.science/r/DARE-anonymous-0C77/maze_datagenerator/README.md**), and removes repeated trajectories to prevent train–dev leakage. A small dataset slice is included for reference.
>
> * MiniBehavior dataset [4]. Based on the official mini_behavior codebase
>  (*https://github.com/StanfordVL/mini_behavior*),
>   we implement a full generator (`generate_ppo_multilevel_dataset.py` in **https://anonymous.4open.science/r/DARE-anonymous-0C77/minibehavior_datagenerator/README.md**).  A PPO agent (Stable-Baselines3) is trained in the INSTALLINGAPRINTER environment for room sizes 7, 8, 9, and 10; for reproducibility, we upload the room-size-7 model used in our experiments. We then extract successful action sequences together with rendered intermediate frames. Both the generator code and a data subset are provided in the anonymous repository.
>
> * FrozenLake dataset [5]. Using the Visual Spatial Planning environment
>   (*https://github.com/UCSB-NLP-Chang/Visual-Spatial-Planning*),
>   we implement `frozen_lake_unified_balance.py`(in **https://anonymous.4open.science/r/DARE-anonymous-0C77/frozenlake_datagenerator/README.md**) to collect Q-table–guided trajectories, following the success/failure balancing rule described in MVoT (50% kept, 50% extended with random actions). Intermediate visual thoughts are rendered in the same format as in MVoT. The generator and a dataset sample are provided.
>
>
> Across all three benchmarks, our reproduced datasets follow the same environment dynamics, trajectory lengths, and intermediate-thought specifications as described in MVoT, ensuring consistency with the original evaluation setting.
>
>
> [1] Li, Chengzu, et al. *Imagine while reasoning in space: Multimodal visualization-of-thought.*(ICML 2025).
>
> [2] Wu, Wenshan, et al. *Mind's eye of LLMs: visualization-of-thought elicits spatial reasoning in large language models.*(NeurIPS 2024).
>
> [3] Ivanitskiy et al. *A configurable library for generating and manipulating maze datasets* (in Hugging Face
>  2023).
>
> [4] Jin et al. *Mini-Behavior: A procedurally generated benchmark for long-horizon decision-making in embodied AI* (NeurIPS workshop 2023).
>
> [5] Wu et al. *VSP: Assessing the dual challenges of perception and reasoning in spatial planning tasks for MLLMs* (ICCV 2024).

---

> ### Author Response · Authors · 2025-11-21
> **Clarifications on Methodological Complexity and Practical Tuning Burden (Q2.1 Response)**
>
> **`Q2.1 Methodological Design`**
>
>  **`A: `** Thanks for your insightful comments. We respectfully clarify that although DARE introduces several conceptual components, the practical tuning burden is low, and in fact substantially lighter than competing efficient-reasoning approaches. Below we highlight the key reasons.
>
> (1) **DARE consists of only *one learned router head per modality* and *a single retention controller***. These components produce token-importance scores and apply pruning according to modality-specific retention targets. Importantly, **DARE introduces no architectural changes to the backbone and requires no multi-stage training**, relying only on a single 10-epoch adaptation. Its core behaviors, including layer-adaptive routing, hop-adaptive routing, and asymmetric vision–text preservation, emerge naturally from this unified learned scoring mechanism, rather than from hand-crafted rules.
>
> (2) **Much of the perceived complexity arises from the paper’s detailed analysis rather than from actual engineering overhead**. In practice, adding DARE requires only inserting two small router heads and enabling pruning during forward passes. There is no modification to the attention kernels, KV-cache layout, or decoding internals.
>
> (3) **The empirical comparisons already demonstrate that DARE is *simpler* and *more stable* than existing SOTA efficient-reasoning methods**.
>
> * *Implicit-efficient baselines such as Heima [6]* require decoding-time implicit schedules, handcrafted step templates, and multi-stage tuning. Despite this higher complexity, Heima underperforms DARE by **17.67%** on MAZE and shows less stable behavior across tasks as detailed in Table A2.
> * *Explicit pruning baselines such as SparseVLM [7]* prune visual tokens under iterative schedules and cannot adapt to the evolving visual–textual interactions across hops. These methods achieve consistently lower accuracy than DARE (e.g., **62.31% vs. 68.09%** on VSR) and often require more FLOPs (e.g., on MAZE: **17,834 G vs. 12,739 G**).
>
> (4) **DARE’s routing mechanism is architecture-agnostic**. We apply the *same formulation, same modules, and same training recipe* to both *Anole-7B and Volcano-7B*, and it also extends naturally to *Chameleon 34B*, without any architecture-specific modifications or tuning heuristics. Hyperparameter robustness is clarified separately in `Q2.2`, where we show that all hyperparameters remain stable and require minimal adjustment.
>
> Table A2: Comparison of Methodological Efficiency Across Baselines
>
> | Method | Core Mechanism | Training / Tuning Complexity | Architecture Modification | Accuracy (MAZE) | FLOPs (MAZE) | Key Observations
> |-|-|-|-|-|-|-
> | DARE (Ours) | Learned token routing via 2 lightweight router heads + retention controller | Single 10-epoch adaptation, no auxiliary losses, no multi-stage or curriculum training; hyperparameters stable (see `Q2.2`) | No modification | 93.32% | 12,739 G | Best accuracy–efficiency trade-off; simple to tune; adapts across layers & hops |
> | Heima (Implicit-Efficient) [6] | Decoding-time implicit schedules + handcrafted templates | High complexity: multi-stage fine-tuning, hand-designed step templates; no supervised token signals → unstable optimization | Requires decoding-time pipelines | 75.65% (−17.67%) | 13,243G | Marginally faster per step (~0.02s), but accuracy significantly lower; tuning harder |
> | SparseVLM (Explicit Vision Pruning) [7] | Iterative visual pruning schedule (uniform across hops) | Low, but cannot adapt to reasoning dynamics; no modality asymmetry | No modification | 87.26% | 17,834 G | Lower accuracy and significantly higher FLOPs; fails to preserve evolving visual–textual cues |
>
> [6] Shen, Xuan, et al. *Efficient reasoning with hidden thinking.* (arXiv 2025).
>
> [7] Zhang, Yuan, et al. *Sparsevlm: Visual token sparsification for efficient vision-language model inference.* (ICML 2025).

---

> ### Author Response · Authors · 2025-11-21
> **Hyperparameter Sensitivity Analysis and Stability Across Tasks (Q2.2 Response)**
>
> **`Q2.2  Hyperparameter Sensitivity`**
>
>  **`A: `** Thank you for highlighting the question of hyperparameter tuning. Although DARE contains multiple interacting components, our experiments show that its practical tuning burden is far smaller than its architecture might suggest. To make this fully transparent, we included a dedicated hyperparameter sensitivity study and an accompanying radar plot (Fig. 7, Lines 513–529 in the revised paper). This figure summarizes the actual measured accuracy deviations across MAZE, MiniBehavior, and VSR when each hyperparameter is perturbed across a broad range.
>
> (1) **Only the target retention ratios for visual(ρᵛ) and textual tokens(ρᵗ) meaningfully influence the quality–efficiency trade-off.** These two parameters control pruning aggressiveness across layers and hops. A small grid search is sufficient to identify good settings, and the optimal values (text 0.7, vision 0.4) transfer across all seven benchmarks and both model families without further tuning.
>
> (2) **All remaining hyperparameters are empirically stable.**  The Gumbel–Softmax temperature varies smoothly, and sweeping it from 0.3 to 1.3 changes accuracy by less than about 1.0–1.3 percentage points across tasks. The pruning threshold shows deviations below roughly 0.2–0.3 percent across wide ranges. The prefix length remains stable for all values greater than or equal to 2, with differences under 0.05 percent. The visual cutoff depth behaves similarly: shifting it by plus or minus three layers changes accuracy by less than 0.1 percent. These results show that DARE’s routing mechanism does not rely on fine-grained calibration and behaves predictably across a wide range of perturbations.
>
> (3) **The default hyperparameters transfer well across architectures (VolCano, Anole-7B) and across both static and dynamic spatial reasoning tasks.** Visual and textual retention remain within narrow, consistent bands across very different problem settings, and the same temperature, threshold, and prefix configurations perform effectively across all evaluated scenarios. This cross-task and cross-architecture consistency indicates that DARE’s hyperparameters are largely architecture-agnostic and domain-agnostic.
>
> A consolidated overview of all hyperparameters, their roles, and their measured sensitivity ranges is provided in Table A3, which summarizes the empirical stability discussed above.
>
> Table A3: Summary of DARE’s Hyperparameters and Sensitivity Observations
> | Category             | Hyperparameter               | Value Used        | Role in DARE                                         | Sensitivity Observation                         |
> |----------------------|-------------------------------|-------------------|-------------------------------------------------------|-------------------------------------------------|
> | Retention targets    | Visual retention ratio ρᵛ    | 0.4              | Controls fraction of visual tokens retained           | Main tuning knob; deviation ≤1.06%              |
> |                      | Text retention ratio ρᵗ      | 0.7              | Controls fraction of textual tokens retained          | Main tuning knob; deviation ≤1.00%              |
> | Routing temperature   | Gumbel–Softmax τ             | 0.7               | Smoothness of differentiable routing scores           | Stable across τ ∈ [0.3, 1.3]; ≤0.8–1.0% impact  |
> | KV-cache routing      | Prefix length κ              | 2                 | Small prefix always preserved for stable decoding     | Stable for κ ≥ 2; ≤0.05% deviation              |
> | Visual pruning        | Visual cutoff depth l_c      | auto-detected     | Switch point from soft to hard visual pruning         | ±3 layer shift changes ≤0.1%                    |
> | Threshold parameters  | Pruning threshold ε          | 0.01              | Threshold for suppressing late-layer visual tokens    | Very robust; ≤0.2–0.3% deviation
>
>
>
> **`Q_typo: L037 MoT should be MVoT`**
>
>  **`A: `** Thank you for pointing out the typo. We have corrected it in the revised version at Line 37.

---

### Meta-Review · Area_Chair_QRBQ · 2026-01-05

**Summary:**

The paper proposes DARE (Dynamic and Asymmetric Routing), a framework designed to enhance the efficiency of multimodal large language models (MLLMs) specifically for multi-hop spatial reasoning tasks. The core innovation lies in a learnable, modality-aware routing mechanism that asymmetrically prunes visual and textual tokens across network layers and reasoning steps. The method also introduces a progressive KV-cache retention policy. Extensive experiments across seven spatial reasoning benchmarks (including dynamic tasks like MAZE and FrozenLake) demonstrate that DARE significantly reduces FLOPs (\~40%) and KV-cache usage (\~46%) while maintaining or improving accuracy compared to state-of-the-art baselines.

The initial reviews were generally positive regarding the motivation and novelty but raised significant concerns regarding reproducibility and methodological complexity.

**Reviewer Concerns:**

Concerns Addressed by Rebuttal:

- Reproducibility of Dynamic Benchmarks (Reviewers UNBZ, ddYw): This was the primary reason for the lower scores (4s). Reviewers noted that the datasets (MAZE, MiniBehavior, FrozenLake) and MVoT baselines were not public, making verification impossible. The authors provided a comprehensive response, releasing an anonymous repository containing full source code, DeepSpeed configurations, and most importantly, the procedural generation scripts to reconstruct the datasets from public environment codebases. They clarified that they reproduced the MVoT baselines from scratch using these specifications.

- Computational Feasibility (Reviewer UNBZ): The reviewer questioned if Anole-7B could actually be trained on 4x A100 40GB GPUs. The authors provided training logs confirming the memory usage fits within this budget using DeepSpeed ZeRO-3 and bfloat16, along with exact wall-clock training times.

- Hyperparameter Sensitivity and Complexity (Reviewers UNBZ, 8DKA): Reviewers worried that the method introduced too many moving parts (retention ratios, thresholds, temperatures) making it hard to tune. The authors added a sensitivity analysis (Radar plot in Fig 7) and new tables (Table A3/A4) demonstrating that the method is robust to hyperparameter variations (except for the global retention ratio, which is the intended control knob). They also clarified that the method is trained end-to-end in a single stage, unlike multi-stage baselines like Heima.

- Flexibility of Reasoning Hops (Reviewer 8DKA): Concerns were raised about whether the model is locked into a fixed number of hops. The authors clarified the "hop" definition and explained that routing is hop-local, allowing the model to handle variable-length trajectories naturally.

Outstanding Concerns:

In my own opinion, there are no major outstanding technical concerns. The reviewers' primary blocking issues were related to the availability of code/data for verification, which the authors have supplied during the rebuttal.

**Reviewer Scores:**

- Reviewer wQMw (Current: 8): This reviewer was already satisfied (noting improvements from a previous submission cycle) and would likely maintain their score of 8.

- Reviewer 8DKA (Current: 6): This reviewer was positive but had questions about applicability and hyperparameters. The detailed response regarding hyperparameter robustness and the explanation of variable-hop reasoning directly addressed their weaknesses. I estimate this reviewer would raise their score to a 7 or 8 given the comprehensive new analysis.

- Reviewer UNBZ (Current: 4): This reviewer liked the design but penalized the paper heavily for reproducibility and doubt regarding computational resources. The authors provided the exact evidence requested (training logs, code, reproducibility scripts). With the "black box" concern removed, this reviewer would likely move to a 6 or 7.

- Reviewer ddYw (Current: 4): This reviewer’s sole listed weakness was the lack of public code/datasets for the dynamic experiments. The authors provided the repository and generation scripts. Consequently, this reviewer would almost certainly change their score to a 6 or higher.

---

### Decision · Program_Chairs · 2026-01-26

Accept (Poster)